# Climate change impacts on floods in West Africa: New insight from two large-scale hydrological models

Serigne Bassirou Diop 1

Job Ekolu 2

Yves Tramblay 3

Bastien Dieppois 2

Stefania Grimaldi 4

Ansoumana Bodian 1

Juliette Blanchet 5

Ponnambalam Rameshwaran 6

Peter Salamon 4

Benjamin Sultan 3

Laboratoire Leïdi "Dynamique des Territoires et Développement", Université Gaston Berger, Saint-Louis, Senegal
Centre for Agroecology, Water and Resilience, Coventry University, Coventry, UK
Espace-Dev, Univ. Montpellier, IRD, Montpellier, France
European Commission, Joint Research Centre (JRC), Ispra, Italy
Univ. Grenoble Alpes, CNRS, IRD, Grenoble INP, IGE, Grenoble, France
UK Centre for Ecology & Hydrology, Wallingford, UK

*Correspondence to: Serigne Bassirou Diop (09.bachir.diop.10@gmail.com)*

# Abstract

West Africa is projected to face unprecedented shifts in temperature and extreme precipitation patterns as a result of climate change. The devastating impacts of river flooding are already being felt in most West African countries, emphasizing the urgent need for comprehensive insights into the frequency and magnitude of floods to guide the design of hydraulic infrastructure for effective flood risk mitigation and water resource management. Despite its significant socio-economic and environmental impacts, flood hazards remain poorly documented in West Africa due to the data-related challenges. This study aims to fill this knowledge gap by providing a large-scale analysis of flood frequency and magnitudes across West Africa, focusing on how climate change may influence future flood trends. To achieve this, we have used two large-scale hydrological models driven by five bias-corrected CMIP6 climate models under two Shared Socioeconomic Pathways (SSPs). The Generalized Extreme Value (GEV) distribution was utilized to analyze trends and detect change points by comparing multiple non-stationary GEV models across historical and future periods for a set of 58 catchments. Both hydrological models consistently projected increases in flood frequency and magnitude across West Africa, despite their differences in hydrological processes representation and calibration schemes.Flood magnitudes are projected to increase at 94% (96%) of stations for the 2-year (20-year) event in the near-term future, and at 88% (93%) of stations for the 2-year (20-year) event in the long-term future, with some locations expected to experience increases exceeding 45%. The findings from this study provide regional-scale insights into the evolving flood risks across West Africa and highlight the urgent need for climate-resilient strategies to safeguard populations and infrastructure against the increasing threat of flood hazards.

**Keywords:** Flood frequency analysis, GEV, GMLE, West Africa, climate change, CMIP, SSP

# 1 Introduction

Anthropogenic changes in atmospheric composition and land use have led to climate change (Houghton et al., 2001; Hansen et al., 2010; Santer et al., 2019; Masson-Delmotte et al., 2021). Climate change, in turn, amplifies the frequency, intensity, and impact of extreme events, such as heatwaves, storms, floods, and droughts at the global scale (IPCC, 2021). West Africa is identified as a hotspot for climate change impacts, as the region is projected to experience unprecedented shifts in both temperature and extreme precipitation patterns (IPCC, 2021). West African populations are therefore becoming increasingly vulnerable for floods and droughts (Tramblay et al., 2020, Rameshwaran et al., 2021). This vulnerability is due to multiple factors such as the region's reliance on rainfed agriculture and the dependence of its rural communities on the natural environment (Krishnamurthy et al., 2012; Totin et al., 2016; Land et al., 2018; Diallo et al., 2020; De Longueville et al., 2020; Matthew et al., 2020). Additionally, the limited economic and institutional resources available to manage and adapt to climate change and natural hazards exacerbate this vulnerability (Roudier et al., 2011; Sultan & Gaetani, 2016; Lalou et al., 2019).

A potential increase in river flooding risks is one of the most frequently studied impacts of climate change (Arnell & Gosling, 2016), because of the devastating economic and environmental impacts it may trigger (EM-DAT, 2015; CRED, 2022; UNDRR, 2023). Such impacts of climate change are already being felt in many West African countries, which experienced several catastrophic floods in the past few years, raising concerns for water management and livelihoods (World Bank, 2021a). It is therefore becoming crucial to develop efficient adaptation strategies for mitigating the adverse effects of flood hazards on West African communities and economies.

Efficient water resources management is essential for sustainable development in West Africa in a changing climate (UNEP, 2020). However, water management requires comprehensive insights into the frequency and magnitude of floods to design appropriate hydraulic infrastructure (Feaster et al., 2023), and quantification of watershed runoff to design reservoirs for agricultural, industrial, and municipal water use (Song et al., 2022). In West Africa however, access to hydrometric data remains a challenge, as the number of stations within hydro-monitoring networks has decreased in recent years (Bodian et al., 2020; Tarpanelli et al., 2023). Existing hydrometric databases, available to estimate design flows, only provide short and often old records (Agoungbome et al., 2018; Tramblay et al., 2021). Therefore, updating these

design flood estimation values (i.e. used to build dams or reservoirs) is essential to ensure that
they accurately represent the current hydroclimatic context of the region (Wasko et al., 2021)

Global Climate Models (GCMs) outputs from the fifth/sixth Coupled Model Intercomparison
Project (CMIP5/6), which contributed to the fifth and sixth Assessment Report (AR5/6) of the
Intergovernmental Panel on Climate Change (IPCC), have provided opportunities to simulate
future hydrological impacts of climate change worldwide. Indeed, CMIP5/6 models use a range
of scenarios that represent different future trajectories to simulate several climate variables,
which help researchers assess the potential long-term impacts of near-term decisions on
emissions reductions and climate policies (Riahi et al., 2017). To understand future trends in
hydrological extremes, climate models are typically used in combination with hydrological
modelling experiments.  However, the simulations from GCMs cannot be used directly to drive
hydrological models as they are associated with systematic biases relative to observational
datasets (Sillmann et al., 2013). Therefore, downscaling and bias-correction algorithms are
routinely applied to leverage the information from GCM outputs (Ehret et al., 2012).
Nevertheless, large uncertainties remain regarding future climate trends in West Africa, partly
due to differences in how climate models simulate projected warming of the North Atlantic and
Mediterranean Sea, affecting the West African Monsoon and projected rainfall changes in the
region (Bichet et al., 2020; IPCC, 2021; Monerie et al., 2023).

As climate change may intensify the hydrological cycle (Gudmundsson et al., 2012),
systematically assessing future flood risks and regional-scale hydrological impacts of future
climate change is crucial for developing effective climate adaptation strategies (Huang et al.,
2024). Due to their simplicity and computational efficiency, lumped hydrological models have
been widely applied in West Africa (Niel et al., 2003; Bodian et al., 2016; 2018; Kwakye &
Bárdossy, 2020; Koubodana et al., 2021). However, because runoff generation is an inherently
spatial and temporally dynamic process, changing environmental conditions may impact flood
frequencies and water availability (Wilson et al., 1979; Haddeland et al., 2002; Descroix et al.,
2018). Although lumped models often perform comparably or even better than distributed
models at the catchment outlet (Reed et al., 2004), their main limitation lies in evaluating the
overall catchment response simply at the outlet, without accounting for the contributions of
upstream individual sub-basins (Cunderlik, 2003; Pokhrel et al., 2008; Jajarmizad et al., 2012).
The main advantage of distributed models is not necessarily a higher accuracy of runoff
simulations at specific points (e.g., outlet or gauge stations), but rather their broader

applicability and ability to simulate the impacts of spatially varying drivers and scenarios (Gebremeskel et al., 2005; Tang et al., 2007; Thielen et al., 2009; Chu et al., 2010; Tran et al., 2018). The interest in large-scale hydrological models has increased due to the need to sustainably manage large river basins and the pervasive global environmental change (Döll et al., 2008). As global hydrological models can capture the variability of hydrological processes across different geographical and climatic contexts, large-scale hydrological modelling has become a key tool for analysing global and regional water resources, assessing climate impacts, and managing water resources (Kauffeldt et al., 2013; Prudhomme et al., 2024). However, running physically based large-scale hydrological models requires numerous input variables that describe the physiographic characteristics of the watersheds (such as soil moisture, land use/land cover, topography, etc.), along with several meteorological forcings. Thus, this complexity limits the widespread use of these models. Brunner et al. (2021) have argued that the limited information on regional flood trends is partly due to the data-related challenges. In the West African context, several studies have shown the increase in extreme rainfall in observations (Taylor et al., 2017, Tramblay et al., 2020, Chagnaud et al., 2022) and future climate scenarios (Dosio et al., 2021, Chagnaud et al., 2023), but very few studies have used GCMs simulations as forcings to drive grid-based large-scale hydrological models to assess the potential impacts of climate change on river flows across West Africa (Rameshwaran et al., 2021; Ekolu et al., 2024, https://africa-hydrology.ceh.ac.uk/). The main objective of this study is to address this gap by assessing the impacts of climate change on floods in the West African region from two large-scale hydrological models driven by data from five bias-corrected CMIP6 GCMs under two Shared Socioeconomic Pathways (SSPs; O'Neill et al., 2017). This article is organised as follows: In Section 2, we describe the study area. Section 3 outlines the materials and methods, including the data used in the analysis, the CMIP6 models and hydrological modelling approach, the non-stationary extreme value analysis framework, and the evaluation of climate change impacts on floods at both local and regional scales. In Section 4, we present and discuss the findings. Finally, main conclusions and perspectives are given in Section 5.

## 2 Materials and Methods

### 2.1 Study area description

West Africa covers about one-fifth of the African continent, extending from the Atlantic coast of Senegal (18°W) to eastern Chad (25°E) and from the Gulf of Guinea (4°N) to the Sahel (25°N) (Figure 1). The region's climate is governed by the Inter-Tropical Convergence Zone (ITCZ) or the Inter-Tropical Discontinuity (ITD), which represents the interface at the ground between moist monsoon air and dry harmattan air with a migratory annual cycle (Pospichal et al., 2010). The West African region features high climatic diversity (Vintrou, 2012), and covers a wide range of ecosystems and bioclimatic regions (Nicholson, 2018). The latitudinal and seasonal oscillation of the ITCZ divides the region into three main climatic domains, namely the Sahel, Sudanian and Guinean zones (Sule & Odekunle, 2016). The Sahel zone is a semi-arid region with a short rainy season and an annual average rainfall not exceeding 600 mm (Figure 1). This domain is highly vulnerable to the adverse effects of climate change (Tian et al., 2023). The Sudanian zone stretches as a broad belt south of the Sahel, receiving an average rainfall of 600 to 1200 mm (Srivast et al., 2023). The Guinean zone, known for its rugged terrain with steep slopes (Orange, 1990), receives abundant rainfall throughout the year, with an annual average between 1200 and 2200 mm (ECOWREX, 2018). These three climate zones are characterized by distinct vegetation (Biaou et al., 2023) and rainy season patterns. The Sahelian and Sudanian domains share a unimodal rainfall pattern, while the Guinean zone experiences a bimodal rainfall pattern of two rainy seasons, driven by the West African Monsoon (Rodríguez-Fonseca et al., 2015; Nicholson, 2018).It is worth noting that nearly half of African watersheds are located in West Africa. The socioeconomic development (agriculture, energy production, and livelihoods) of the region relies highly on the water resources provided by these transboundary basins and aquifers (World Bank, 2021b).

## 2.2 Observational data and climate forcings for hydrological experiments

Daily streamflow data for the period 1950-2018 were obtained from the African Database of Hydrometric Indices (ADHI) recently developed by Tramblay et al. (2021). This database provides hydrometric indices computed from different data sources, with daily discharge time series that span at least 10 years. In the ADHI database, the size of the 441 West African catchments ranges from 95 to 2,150,000 km2, and some stations have daily discharge data spanning over 44 years. Figure 1 shows the spatial distribution of the ADHI stations used in this study, and Supplementary Table S1 gives information on their geographical locations (longitude and latitude), catchment areas, mean annual catchment-averaged rainfall, mean

annual streamflow, and the range of years over which streamflow data is available. We only
selected watersheds from the ADHI database that met the following three criteria: (i) low
regulation, determined through visual inspection of dam locations relative to watershed outlets
(see Supplementary Figure S1), combined with a year-by-year analysis of annual hydrographs
to assess the impact of dam operations on streamflow, (ii) surface area of less than 150,000
km², and (iii) a daily streamflow time series covering a minimum of 10 years between the 1950
and 2018. To address the challenges associated with missing data in the database, we conducted
a visual inspection of hydrographs at each station as illustrated by Supplementary Figure S2.
Years with data gaps near the flood peak were excluded from the analysis to avoid the risk of
missing the true annual peak flood (Wilcox et al., 2018). Through this careful screening
process, we ensured that no AMF values were derived from periods characterized by  a lot of
missing data. It is important to note that the observational streamflow data are not used to
calibrate or drive the hydrological models. Instead, these observations serve as an independent
benchmark to evaluate the ability of the hydrological models to reproduce key flood statistics
during the historical period. The LISFLOOD model was calibrated using the ERA5 reanalysis
dataset, which provides consistent and high-resolution precipitation and temperature fields.
Moreover, ERA5 was also used as a reference for the bias correction of the five climate models
from the CMIP6 ensemble that were used to drive the hydrological simulations for both the
historical and future periods (see Section 2.4).

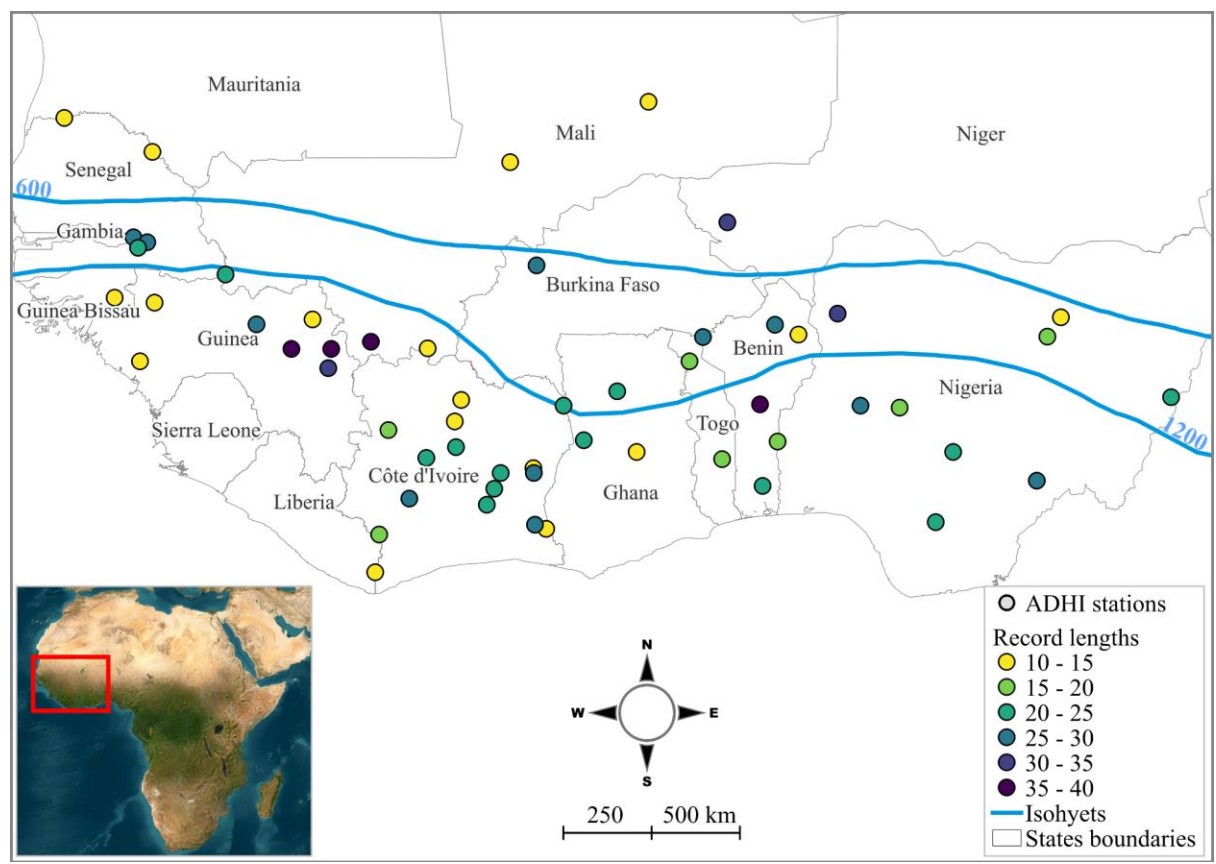

Figure 1: Spatial distribution of the stations used in this study, covering the three climatic zones in the West African region, as delimited by the blue isohyets (600 mm and 1200 mm annual rainfall) on the map. The color of the circles indicates the record lengths of flood data (in years). The blue lines represent isohyets delimiting West African climatic regions, and the grey lines indicate the borders of West African countries (African continent map from ©NASA 2005).

## 2.3 hydrological models

Two grid-based large-scale hydrological models were used to simulate river flows for the period from 1950 to 2010: the HMF-WA model (the Hydrological Modelling Framework for West Africa; Rameshwaran et al., 2021) and the Open Source (OS) LISFLOOD model (Van Der Knijff et al., 2010), thereafter referred to as LISFLOOD.  The HMF-WA model is adapted from the modular HMF model, and is designed for large-scale applications across West Africa (Rameshwaran et al., 2021). It employs a vertically integrated soil moisture scheme to simulate runoff production, driven by rainfall and potential evaporation inputs. Runoff generation considers soil drainage and a spatial probability distribution of soil moisture. Routing is based on a kinematic wave approach (Bell et al., 2007), with parallel pathways for surface and subsurface flow. Key enhancements over the classical HMF model include modules to simulate

wetland inundation, endorheic basins, and anthropogenic water withdrawals, making it well-
suited for semi-arid environments with complex hydrology (Rameshwaran et al., 2021). HMF-
WA simulates spatially consistent river flows across West Africa at a $0.1° \times 0.1°$ spatial
resolution. Although it has not yet been specifically calibrated to individual West African
catchments using observed streamflow data where the model hydrology is configured to local
conditions using spatial datasets of physical and soil properties, HMF-WA model evaluation
against observational data indicates that it performs reasonably well in simulating both daily
high and low river flows across most catchments. The median values of NSE (Nash-Sutcliffe
efficiency), NSElog, and  BIAS are 0.62, 0.82, and 0.06 (6 %), respectively (Rameshwaran et
al., 2021).

The LISFLOOD model, developed by the Joint Research Centre (JRC) of the European
Commission (https://ec-jrc.github.io/lisflood/), is a physical, spatially distributed hydrological
model, designed for simulating several hydrological processes that occur in a catchment (Van
Der Knijff et al., 2010). The LISFLOOD model simulates water processes using a three-layer
soil water balance, along with groundwater and subsurface flow models. It accounts for several
processes such as snow accumulation/melt, infiltration, evapotranspiration, groundwater flow,
surface runoff, etc. Moreover, it supports the integration of human influences such as reservoirs
and water abstraction. The numerical LISFLOOD simulation is driven by meteorological
forcing (precipitation, temperature, and evapotranspiration) combined with high-resolution
spatial data on terrain morphology, soil characteristics, land use, and water demand. This
integrated setup allows the model to simulate runoff processes under diverse climatic and
socio-economic conditions, capturing both natural and anthropogenic influences across
heterogeneous landscapes. The runoff produced at every grid cell within the model domain is
routed through the river network using a kinematic wave approach. The LISFLOOD version
used in this study (OS LISFLOOD v4.1.3) was calibrated with a 0.05° (~5 km) resolution in
its quasi-global implementation covering a longitude range from -180° to 180° and a latitude
range from 90° to -60°, using in-situ discharge gauge stations with at least four years of daily
measurements recorded after 1 January 1982. In this setup, model parameters are linked to
global geospatial datasets describing catchment morphology and river networks, land use,
vegetation characteristics, soil properties, lake distribution, and water demand (Salamon et al.,
2024; Choulga et al., 2024). The Distributed Evolutionary Algorithms in Python (DEAP; Fortin
et al., 2012) framework was applied to optimize parameters in gauged catchments, with the
modified Kling-Gupta Efficiency (KGE; Gupta et al., 2009) utilized as the objective function.
Calibration was performed over a continuous simulation period using ERA5 reanalysis
meteorological forcing. Due to the varying length and temporal coverage of the discharge
records used for calibration, model performance was assessed using all available observational
data at each station, rather than splitting the records into separate calibration and validation
periods. The LISFLOOD calibration tool is freely available at https://github.com/ec-
jrc/lisflood-calibration.

Globally, while both models use a kinematic wave routing scheme, HMF-WA and LISFLOOD
differ significantly in their hydrological process representation. HMF-WA applies a vertically
integrated soil moisture scheme with simplified runoff generation based on spatial soil moisture
distribution. In contrast, LISFLOOD features a more detailed, physically-based three-layer soil
model with an explicit representation of groundwater, snow processes, and anthropogenic
influences. Furthermore, LISFLOOD has been calibrated using in-situ discharge data.
Nevertheless, while calibration can enhance the accuracy of discharge simulations, several
studies have highlighted that uncalibrated global hydrological models often exhibit comparable
sensitivity to climate variability as the regional calibrated hydrological models, particularly
when assessing relative changes in extreme events between future and historical periods
(Gosling et al., 2017; Zhao et al., 2025). Therefore, whether a calibrated hydrological model
offers different climate change projections than an uncalibrated model needs further
investigation (Pechlivanidis et al., 2017).

## 2.4 Bias-corrected CMIP6 models and scenarios

The sixth phase of the Coupled Model Intercomparison Project (CMIP6) provides simulations
from GCMs for the preindustrial period (1850–2014) and future climate projections (2015–
2100) (Noël et al., 2022). To assess future climate impacts on floods, we have used five (5)
daily GCMs rainfall and temperature outputs from the CMIP6 experiments (https://esgf-
node.llnl.gov/search/cmip6). Table 1 gives the institute name and references of the CMIP6
climate models used in this study. These GCMS encompass a range of climate sensitivities,
with Equilibrium Climate Sensitivity (ECS) values ranging from 2.98 to 5.34 (IPCC, 2021).
The GCMs were selected based on their availability for the study area. Due to their
accessibility, these GCMs have been widely used for climate impact assessments in Africa
(Dosio et al., 2019; Almazroui et al., 2020; Klutse et al., 2021; Babaousmail et al., 2023; Nooni
et al., 2023). The Cumulative Distribution Function-transform (CDF-t) (Michelangeli et al.,

2009) was used to bias-correct the GCMs outputs. The CDF-t approach involves mapping the cumulative distribution function (CDF) from a GCM in the historical period to the observed CDF, then applying the same mapping to the GCM's future CDF (Flaounas et al., 2013; Pierce et al., 2015; Famien et al., 2018). The CDF-t method requires high-resolution observational data to work properly. The EWEMBI dataset (E2OBS, WFDEI, and ERA-I data, bias-corrected for ISIMIP; Frieler et al., 2017; Lange, 2018, 2019) was used to bias-correct the climate variables to drive the HMF-WA hydrological model. Similarly, the ERA5-land reanalysis (Muñoz-Sabater et al., 2021) was used for bias-correcting the GCMs outputs for the LISFLOOD model. The EWEMBI dataset was developed to support bias correction of climate input data used in impact assessments in phase 2b of the Inter-Sectoral Impact Model Intercomparison Project (ISIMIP2b; Frieler et al., 2017). EWEMBI dataset (https://dataservices.gfz-potsdam.de/pik/showshort.php?id=escidoc:3928916) provides global spatial coverage with 0.5° x 0.5° spatial and daily temporal resolutions. It integrates multiple sources, including ERA-Interim reanalysis data (Dee et al., 2011), the WATCH Forcing Data methodology applied to ERA-Interim (WFDEI; Weedon et al., 2014), the eartH2Observe forcing dataset (E2OBS; Calton et al., 2016), and the NASA/GEWEX Surface Radiation Budget data (SRB; Stackhouse Jr. et al., 2011). Meanwhile, the ERA5 dataset is a global atmospheric reanalysis product developed by the Copernicus Climate Change Service (C3S) at ECMWF (European Centre for Medium-Range Weather Forecasts ReAnalysis). It is the fifth generation of atmospheric reanalysis based on 4D-Var (four-dimensional variational) data assimilation using Cycle 41r2 of the ECMWF Integrated Forecasting System (IFS) (Hersbach et al., 2020). ERA5 replaces the now outdated ERA-Interim reanalysis (Dee et al., 2011), and provides global spatial coverage from 1979 until the present, with a finer spatial and temporal resolution of 0.25° x 0.25° and 1 hour, respectively. The bias-corrected simulations are post-processed onto the 0.1° x 0.1° (~10 km x 10 km) HMF-WA model grid (Rameshwaran et al., 2021, 2022), and onto the 0.05° x 0.05° (~5 km x 5 km) LISFLOOD model grid for the period 1950-2100. CMIP6 models use five Shared Socioeconomic Pathways (SSPs). SSPs are an updated framework of climate scenarios, building upon the CMIP5 Representative Concentration Pathways (RCPs) while maintaining consistency in the 2100 radiative forcing levels. SSPs describe the socioeconomic factors (population growth, economic development, technological advancements, and governance) which can influence greenhouse gas emissions and adaptation strategies (O'Neill et al., 2017). Two Shared Socioeconomic Pathways (SSPs) are analysed in this study: the SSP2-4.5 (Middle of the Road) and the SSP5-8.5 (Fossil-Fueled Development). Rather than including the full range of SSPs, we focus on SSP2-4.5 and SSP5-

8.5 narratives, which represent moderate and high emission trajectories, respectively. SSP2-4.5 is considered as a "middle-of-the-road" scenario, that is consistent with current national policies and moderate progress towards emission reduction commitments (). In contrast, SSP5-8.5 represents a high emissions pathway, allowing us to explore the upper limits of potential impacts under continued fossil fuel dependence and minimal climate policy intervention. While SSP5-8.5 has been criticized as an "overly pessimistic" narrative (Pielke & Ritchie, 2021), it remains widely used in climate impact assessments to evaluate the vulnerability of socio-environmental systems under a "no-climate policy" world.

**Table 1:** Bias-corrected CMIP6 climate models used in this study

| Institute | Climate Model | References |
|---|---|---|
| Max Planck Institute for Meteorology (Germany) | MPI-ESM1-2-HR | (Mauritsen et al., 2019) |
| Meteorological Research Institute (Japan) | MRI-ESM2-0 | (Yukimoto et al., 2019) |
| Institute Pierre-Simon Laplace (France) | IPSL-CM6A-LR | (Boucher et al., 2020) |
| Met Office Hadley Centre (UK) | UKESM1-0-LL | (Mulcahy et al., 2020) |
| Geophysical Fluid Dynamics Laboratory (USA) | GFDL-ESM4 | (Dunne et al., 2020) |

## 2.5 Evaluation of hydrological models

The two hydrological models are evaluated over the period 1950-2014, which represents a compromise between the period covered by the ADHI database and the historical CMIP6 GCM simulations. To achieve this, we use the two-sample Anderson-Darling (AD) test at the 0.05 significance level (Scholz & Stephens, 1986) to compare the distributions of extreme values observed and simulated by the hydrological models. The null hypothesis of the AD test assumes that the simulated and observed AMF follow the same statistical distribution. The Block-Maxima approach (Gumbel, 1958) is used to construct extreme value time series, by extracting the annual maximum flow (AMF) from the daily discharge time series over the period 1950-2014. Unlike the Kolmogorov-Smirnov (KS) test (Berger & Zhou, 2014), which measures the maximum distance between two cumulative distribution functions (CDFs), the AD test assesses the overall distance between these CDFs, giving more weight to the tails of distributions. As a result, the AD test is more sensitive than the KS test in the tails of distributions and is therefore more suitable for comparing extreme values distributions (Engmann & Cousineau, 2011). That

said, the AD test also has a limitation as the reliability of an empirical CDF can be affected by
small sample sizes, particularly in the tails of the distribution. The performance of each
hydrological model is given here by the proportion of CMIP6 simulations (among the 5) for
which the AD test has failed. It is important to note that the AD test is only used herein to
assess regional-scale performance of hydrological models, and not as a filtering criterion for
inclusion or exclusion of models or stations.

## 2.6 Extremes Values Analysis Framework
### 2.6.1 The Generalized Extreme Value Distribution
According to the theory of extreme values, based on the Fisher–Tippett theorem, the
Generalized Extreme Value (GEV) is the limiting distribution of independent and identically
distributed random variables (Coles, 2001). The GEV is among the most frequently used
distributions for extreme value analysis. It is a continuous three-parameter distribution that can
account for non-stationarity, which refers to changes in statistical properties over time. This is
achieved by allowing the parameters to vary as a function of time or other covariates (Hamdi
et al., 2018; Wilcox et al., 2018). We, therefore, used the GEV to model the AMF series from
each hydrological model simulations forced with the five CMIP6 climate models at each
catchment. There are three parameters (location, scale and shape) in the GEV distribution
(Hossain et al., 2021). In flood frequency analysis, each GEV parameter plays a distinct role in
understanding and projecting flood behaviour (Lawrence, 2020; Wasko et al, 2021). The
location parameter ($\mu$) indicates the central tendency of flood magnitudes, with higher values
suggesting a shift towards more frequent or severe floods. The scale parameter ($\sigma$) measures
the variability or dispersion of the distribution, with larger values indicating greater uncertainty
and a broader range of flood magnitudes. The shape parameter ($\xi$) governs the tail behaviour
of the GEV distribution, which encompasses three types of extreme value distributions (Coles,
2001): (i) a positive  shape parameter ($\xi > 0$) indicates a heavy-tailed Fréchet case (Fréchet,
1927), suggesting an increased probability of extreme flooding events, (ii) a null shape
parameter ($\xi = 0$) suggests a light-tailed Gumbel class (Gumbel, 1958), and (iii) a negative
shape parameter ($\xi < 0$) indicates a short-tailed or (bounded) negative-Weibull distribution
(Weibull, 1951). This parameter is crucial for assessing the risk of rare floods and informing
the design infrastructure to withstand such extremes. Equation (1) presents the cumulative
distribution function (CDF) of the GEV (Coles, 2001).

$$F(x; u, \alpha, \xi) = exp\left\{-\left[1 - \xi\frac{(x-u)}{\alpha}\right]^{1/\xi}\right\} \quad \kappa \neq 0$$

$$F(x; \xi, \alpha) = exp\left\{-exp\left[-\frac{(x-u)}{\alpha}\right]\right\} \quad \kappa = 0$$

(1)

Where $x$, $u$, $\alpha$, et $\xi$ are the data, location, scale, and shape parameters respectively, and $(u + \alpha/\xi) \leq x < \infty$ if $\xi < 0$ ; $-\infty < x < \infty$ if $\xi = 0$ ; $-\infty < x \leq (u + \alpha/\xi)$ if $\kappa > 0$.

Efficiently estimating the GEV parameters is crucial for the precise characterization and analysis of extreme events (Rai et al., 2024). We have used the Generalized (Penalized) Maximum Likelihood Estimation (GMLE) method (Martins & Stedinger, 2000) to estimate the GEV parameters in a non-stationary context, by allowing the model parameters to vary with time (Coles, 2001). The GMLE method overcomes the limitations of the well-known MLE (Fisher, 1992) method for small sample size (Hossain et al., 2021). To achieve this, Martins & Stedinger (2000) used a beta distribution (with shape parameters p = 6 and q = 9) as a prior to constraint the values of the GEV shape parameter in the interval [-0.5, +0.5], avoiding large negative values of the shape parameter. This approach has been used in several studies to estimate the GEV parameters in both stationary and non-stationary contexts (El Adlouni et al., 2007; Panthou et al., 2013; Tramblay et al., 2024). However, the original prior distribution from Martins & Stedinger (2000) is not well-suited for West Africa, as it results in shape parameter estimates below -0.5 for several stations, as illustrated in Supplementary Figure S3. Here, we therefore use a normal distribution as a prior for the GMLE method. This normal distribution is fitted to the GEV shape parameter values estimated on 98 AMF series spanning a minimum of 20 years over the period 1950-2018 from the ADHI database Tramblay et al. (2021) using the L-moments method (Hosking, 1990). The newly developed regional prior, modelled as a normal distribution, has a mean of -0.24 and a standard deviation of 0.16 (see Supplementary Figure S3), and is used to fit the GEV distribution to the historical and projected annual peak flood time series generated by hydrological models driven by the CMIP6 GCMs.

## 2.6.2 Determining magnitude and direction of changes in flood events

To analyse future changes in floods, we compare two 30-year future periods (a near-term future [2031–2060] and a long-term future [2071–2100]) to a reference historical period (1985-2014) at stations where there is a good fit between observed (OBS) AMF series and hydrological

models simulations (HIST) according to the Anderson-Darling (AD) test (at 0.05 level), and also in stations at which the null hypothesis of the AD test is rejected. We have chosen to work with the 2-year and 20-year floods to analyse the impacts of climate change in West Africa. The 2-year return period indicates relatively frequent flood events, and this information is essential for understanding and managing risks associated with flooding. The 20-year flood event is frequently used for comparative purposes in various studies, as it balances the rarity of extreme events (data length limitations) and the uncertainty in the estimated return levels (Dawson et al., 2005; Tramblay & Somot, 2018; Han et al., 2022). Thus, the 2- and 20-year flood quantiles are computed at each station for the three 30-year periods using the GEV model fitted to the AMF series by the GMLE method. Changes in flood are quantified in this study by computing the ratio of the difference between the future flood quantile (Qfuture) and the historical flood quantile (Qhist) to Qhist itself. To assess the statistical significance of the differences between the historical and future flood quantiles, we have used the parametric bootstrapping approach. After estimating the GEV distribution parameters, we have generated 2500 simulations of annual peak floods for each subperiod (with each simulation representing a sample of 30 data points). We have then recomputed the 2-year and 20-year flood quantiles for each simulation. The significance of the differences between the quantiles was evaluated at the 0.05 level. It is crucial to consider the degree of consensus among multiple climate models to reduce the potential noise in the projections and reach robust conclusions (Awotwi et al., 2021; Dosio et al., 2021)**.** Here we have computed a multi-model index of agreement (MIA) as introduced by Tramblay & Somot (2018), to present the results in terms of the proportion of CMIP6 models projecting significant change for each station. The MIA allows the assessment of the robustness of climate model projections, ensuring cross-catchment comparability due to its standardised scale ranging from -1 to 1, according to the direction of change (i.e., MIA = 1 (-1) if all models project an increasing (decreasing) trend).

$$MIA = \frac{1}{n}\left(\sum_{m=1}^{n} i_m\right) \qquad (2)$$

From equation (6), for a given CMIP6 model *(m)*, $i_m = 1$ for regionally significant upward trends, $i_m = -1$ for significant negative trends, and $i_m = 0$ when no significant trends are detected, across *n* climate simulations.

### 2.6.3 Determining temporal functions for GEV parameters and modelling of non-stationary extreme values

While the previous section focused on the magnitude and direction of changes in flood events under different scenarios, this section describes the methodology used to identify when these changes began. Understanding how the parameters of the GEV distribution might shift under future climate scenarios is a critical question that needs to be addressed given the accelerating impacts of global warming on environmental conditions. Answering this question can inform a more reliable modelling process to estimate flood quantiles. Several studies have suggested that both the location and scale parameters of the GEV distribution should be adjusted proportionally to account for the effects of climate change (Stedinger & Griffis, 2011; Prosdocimi & Kjeldsen, 2021; Jayaweera et al., 2024). Here, to determine the appropriate temporal function for the non-stationary GEV, the trends in GEV parameters are detected using the non-parametric Mann-Kendall test (Mann, 1945; Kendall, 1975). As the test is applied to parameters estimated over moving windows, it is important to note that temporal correlation is introduced, which can bias the results of the original Mann-Kendall test, as it assumes independence of observations. To address this, we have applied a modified version of the test based on the Hamed & Rao (1998) variance correction approach, specifically adapted for serially correlated data. A window size of 30 years has been selected to ensure sufficient data to fit the stationary GEV model (SGEV), with a total of 121 windows. For each window, each hydrological model (LISFLOOD and HMF-WA) and each climate scenario (SSP2-4.5 and SSP5.8-5), the SGEV is fitted to AMF series from the averaged hydrological simulations driven by data from the CMIP6 models. The Mann-Kendall test is then applied to the series of estimated parameters at the 0.05 significance level.

Based on the results of the trend analysis of the GEV parameters, the location ($\mu$) and scale ($\sigma$) parameters are expressed as linear functions of time, denoted as $\mu(t)$ and $\sigma(t)$, while the shape parameter remains constant. Thus, the non-stationary GEV model involves a vector $\psi=[\mu_0;\mu_1;\sigma_0:\sigma_1:\xi]$ of five unknown parameters. We have decided to keep the shape parameters constant because it is uncommon for researchers to model all three GEV parameters as covariate-dependent functions. Indeed, adding this level of complexity can significantly complicate the model parameters estimation, particularly the shape parameter (Katz, 2013; Papalexiou & Koutsoyiannis, 2013). Allowing any starting date (year $t_0$) of a possible significant trend in the GEV location and scale parameter, we have considered three cases of the non-stationary GEV (NSGEV; cf. Equations 3-5):

● Case 1 (GEV1): a linear trend with no breakpoint (i.e., a single trend over the entire
record for both the location and scale parameters):

$$\mu(t) = \mu_0 + \mu_1 t \; ; \; \sigma(t) = \sigma_0 + \sigma_1 t \qquad \text{for} \quad t \leq t_0 \qquad (3)$$

● Case 2 (GEV2): a linear trend after a breakpoint (i.e., the location and scale parameters
are constant before the year $t_0$ and linearly dependent on time after $t_0$):

$$\mu(t) = \mu_0 \; ; \; \sigma(t) = \sigma_0 \qquad \text{for} \quad t \leq t_0$$
$$\mu(t) = \mu_0 + \mu_1(t\text{-}t_0) \; ; \; \sigma(t) = \sigma_0 + \sigma_1(t\text{-}t_0) \qquad \text{for} \quad t \geq t_0 \qquad (4)$$

● Case 3 (GEV3): both trends before and after a breakpoint are considered (i.e., a linear
trend before and after year $t_0$ for both location and scale parameters):

$$\mu(t) = \mu_0 + \mu_1(t_0\text{-}t) \; ; \; \sigma(t) = \sigma_0 + \sigma_1(t_0\text{-}t) \qquad \text{for} \quad t \leq t_0$$
$$\mu(t) = \mu_0 + \mu_1(t\text{-}t_0) \; ; \; \sigma(t) = \sigma_0 + \sigma_1(t\text{-}t_0) \qquad \text{for} \quad t \geq t_0 \qquad (5)$$

Unlike in Wilcox et al. (2018), where breakpoints are defined independently for $\mu(t)$ and $\sigma(t)$,
in the present study, we assume a common breakpoint for both parameters. This means that
both $\mu(t)$ and $\sigma(t)$ change simultaneously at the same point in time. To ensure that the NSGEV
model is fitted with sufficient data, the first start year is set no earlier than 20 years after the
beginning of the time series (1950) and the last start year is set no later than 20 years before
the end of the time series (2100). Thus, the possible starting years of change ($t_0$) fall between
1970 and 2070. There are as many NSGEV models as there are breakpoints or starting years,
and the non-stationary model with the highest log-likelihood is selected (see Supplementary
Figure S4). The procedure described above is inspired by several studies that focused on
detecting trends in hydroclimatic time series using non-stationary GEV (Hawkins & Sutton,
2012; Panthou et al., 2013; Blanchet et al., 2018; Hamdi et al., 2018; Tramblay & Somot, 2018;
Wilcox et al., 2018).
Once the best breakpoint has been determined for each time-varying GEV model based on the
log-likelihood profile, the trend models (GEV1, GEV2 and GEV3) are compared with each
other using the Akaike information criterion (AIC; Akaike, 1974). The AIC criterion is widely
used to compare multiple statistical models by assessing their goodness-of-fit. It accounts for
the trade-off between a model's fit to the data and its complexity, by penalising for more
complex models. While a more complex model may provide a better fit, it often does not
provide sufficient improvement to justify the addition of extra parameters (Wilcox et al., 2018).
Thus, the AIC is well-suited for evaluating the performance of non-stationary GEV models.
Furthermore, a deviance test (D) based on likelihood ratio (LR; Coles, 2001) is performed at
the 0.05 significance level between the best GEV trend model selected previously based on the
AIC criterion and the stationary GEV model (SGEV). The LR test allows us to determine the
best model between two competing nested models by comparing the D-statistic given by
Equation (6) to the chi-square ($x^2$) distribution.
$$D = 2\{\log(ML_{NSGEV}) - \log(ML_{SGEV})\} \qquad (6)$$
From Equation (6), D represents the deviance test statistic value (referred to as D-statistic
above), $\log(ML_{NSGEV})$ and $\log(ML_{SGEV})$ are the maximised log-likelihood functions of the
NSGEV and the SGEV, respectively. Letting $c_\alpha$ be the $(1 - \alpha)$ quantile of the chi-square
distribution (where $\alpha$ represents the level of significance), with $\upsilon$ degrees of freedom equal
to the difference in the number of model parameters between the non-stationary and
stationary models, the non-stationary GEV is accepted at the level $\alpha$ if the D-statistic is greater
than $c_\alpha$, meaning a significant trend in the data.
The null hypothesis of the deviance test assumes that the stationary GEV model provides a
better fit to the data than the non-stationary model, indicating that there is no significant trend
in the AMF. However, the presence of spatial cross-correlations across stations may bias the
results of simultaneous multiple local tests by increasing the likelihood of detecting false
positives (Farris et al., 2021). To assess the field significance of local trends detected in AMF
series in the study area, we implement the False Discovery Rate (FDR) procedure (Hochberg
& Benjamini, 1995). The FDR's null hypothesis assumes that none of the stations across the
region exhibits a significant trend in AMF (i.e., all local null hypotheses are actually true). The
FDR aims to reduce Type 1 errors (Mudge et al., 2012), by adjusting the vector of p-values
from the set of at-site tests (Wilks, 2006). Due to its advantages over other methods, such as
dealing with spatial autocorrelation, the FDR approach has been used in many studies of
hydroclimatic variables (Khaliq et al., 2009). For consistency with local deviance and MK tests,
the FDR procedure is computed at 0.05 global significance level ($\alpha$global). The FDR test rejects
the local null hypothesis when the corresponding FDR-adjusted p-value is lower than $\alpha$global.
Field significance is declared if the local null hypothesis is rejected at least once within the
study area (Wilks, 2016).

## 3 Results and discussions

### 3.1 Assessing the performance of hydrological models

The two hydrological models' performance is assessed over the period 1950-2014 by applying the two-sample Anderson-Darling (AD). The results of the statistical evaluation of the two hydrological models are shown in Figure 2. The performance of each model at each station is assessed based on the proportion of CMIP6 models that fail the Anderson-Darling test at the 0.05 significance level. Specifically, if more than two out of five CMIP6 simulations fail the test at a given station, the hydrological model is considered to perform poorly at that station. Considering this evaluation criterion, the LISFLOOD hydrological model performs well at 64 % of the stations, while the HMF-WA model performs satisfactorily at only 24 % of the stations (Figure 2). Although both models are semi-physically based and spatially distributed, the LISFLOOD model outperforms the HMF-WA model in simulating extreme flows in West Africa (Figure 2). These findings are consistent with those of Ekolu et al. (2025), who reported that the LISFLOOD model effectively simulates the hydrological cycle and captures the specific characteristics of hydrological droughts and floods in West Africa. This difference in performance can be attributed to several factors: (i) the LISFLOOD model was run at a finer resolution (0.05° x 0.05°) compared to the coarser resolution of 0.1° x 0.1° used by the HMF-WA model (Rameshwaran et al., 2021); (ii) the HMF-WA model includes fewer meteorological forcings and only a limited number of hydrological processes (specifically wetlands, anthropogenic water use, and endorheic rivers), whereas the LISFLOOD model can incorporate over 70 different processes depending on the target application (i.e., rainfall-runoff transformation, flood and drought forecasting) and the required level of configuration (more detailed information on the configuration of LISFLOOD can be found at https://ec-jrc.github.io/lisflood-model; and (iii) the HMF-WA model has not been calibrated to individual west African catchment conditions with observed flow data, and its performance depends on the accuracy of spatial datasets of physical and soil properties (e.g., wetlands, anthropogenic water use, and endorheic rivers) used to configure the model's hydrology to local conditions (Rameshwaran et al., 2021). In contrast, the LISFLOOD model has been regionally calibrated using in-situ discharge observations, with discharge time series spanning at least four years after 01 January 1982. Consequently, while the distributed nature of the HMF-WA model aims to improve the understanding of regional climate change impacts in a spatially coherent manner

across West Africa, it does not necessarily lead to better modelling of extreme flows in the
various climates and socioeconomic contexts of the region without calibration. Runoff
generation is inherently a spatially distributed process. As such, the spatial resolution of a
distributed hydrological model can significantly affect its ability to capture spatial variability
of key watershed characteristics, such as topographic features, land cover heterogeneity, and
precipitation gradients (Wolock & Price, 1994; Haddeland et al., 2002). A coarser spatial
resolution limits the level of detail that can be represented in hydrological simulations,
potentially overlooking important small-scale processes. Furthermore, as hydrological models
are simplified representations of complex watershed processes, a calibration phase is often
necessary to compensate for limited information on spatial variability of physiographical and
meteorological catchments attributes, and to improve model performance in simulating the
watershed's hydrological cycle (Bruneau et al., 1995). However, many river basins in West
Africa have a limited number of in situ observational networks to provide the current state of
hydrological information (Ndehedehe, 2019). This limits the optimal parameterization of large-
scale hydrological models and may introduce uncertainties in model outputs. In addition, the
satisfactory performance of the LISFLOOD model indicates that, although a flood-centered
calibration approach could potentially improve its ability to capture extreme flows and their
trends (Wasko et al., 2021), the current model setup provides a satisfactory basis for regional-
scale flood trend assessments.

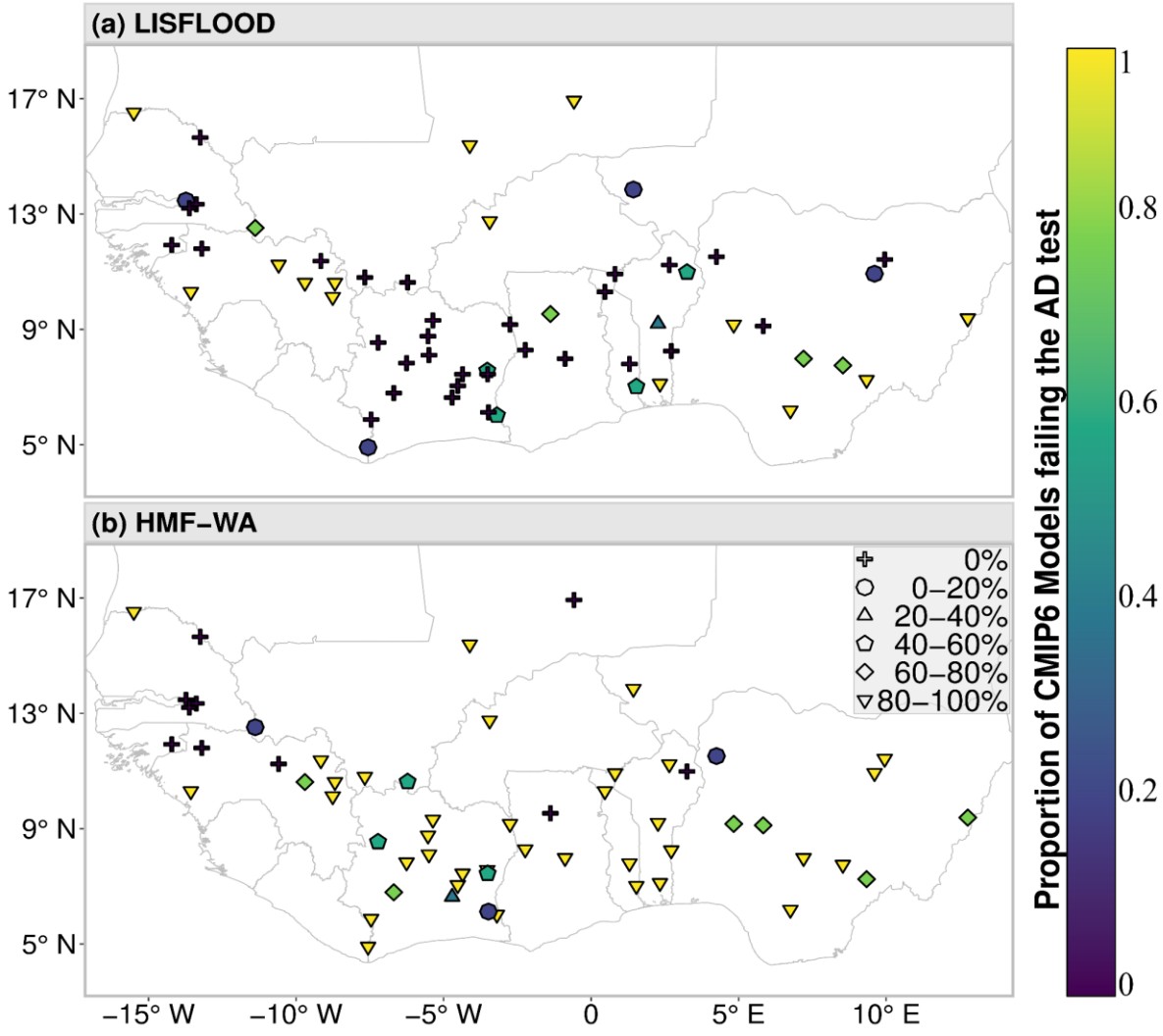

Figure 2: Statistical evaluation of the two hydrological models: a) Two-sample Anderson-Darling (AD) goodness-of-fit (GOF) test at 0.05 statistical significance level at each station between the AMF of daily OBS from the ADHI database and annual maxima flow of HIST from LISFLOOD daily simulations forced with the five CMIP6 GCMs (GFDL, IPSL, MPI, MRI, and UKESM) over the period 1950-2014. b) same as a) but using HMF-WA as hydrological model. The fill color of the markers indicates the proportion of CMIP6 models (out of five) for which the AD test null hypothesis (i.e., simulated and observed AMF follow the same statistical distribution) is rejected at the 0.05 significance level. Marker shapes correspond to binned categories of this proportion, as indicated in the legend.

To further assess the performance of the hydrological models in capturing extreme flows, we computed the Relative Bias between the AMF simulated by the LISFLOOD-CMIP6 and HMF-WA-CMIP6 hydrological models and the observed AMF from the ADHI database. This comparison was performed over the historical period (1950–2014), focusing on the

climatological characteristics of AMF (median values) rather than on year-to-year correspondence. This approach allows us to evaluate whether the hydrological models tend to overestimate or underestimate flood peaks, considering climate models individually. As shown in Figure 3, the HMF-WA model consistently shows a negative relative bias across all GCMs, with median values ranging from -52 % (IPSL) to -46 % (UKESM) across the region. These negative biases suggest a tendency of the HMF-WA model to underestimate peak flow. The LISFLOOD model, in contrast, shows lower bias than the HMF-WA model, with a mix of slight underestimations and even overestimations (Figure 3). For instance, the median values for the LISFLOOD model simulations range from -14 % (MPI) to 7 % (GFDL). Although the LISFLOOD model also shows negative biases with most GCMs, such as IPSL, MPI, MRI, and UKESM, the magnitude of these biases is much smaller compared to the HMF-WA model. Nevertheless, whether a calibrated hydrological model offers more reliable climate change projections than an uncalibrated model, which may perform less accurately in reproducing historical conditions (Pechlivanidis et al., 2017), remains questionable. Examining whether their capacity to simulate hydrological responses to historical climate is influencing projected trends for climate change impacts remain important, especially considering that most projections of climate change impacts on African hydrological trends were produced using uncalibrated models (Davie et al., 2013; Sauer et al., 2021).

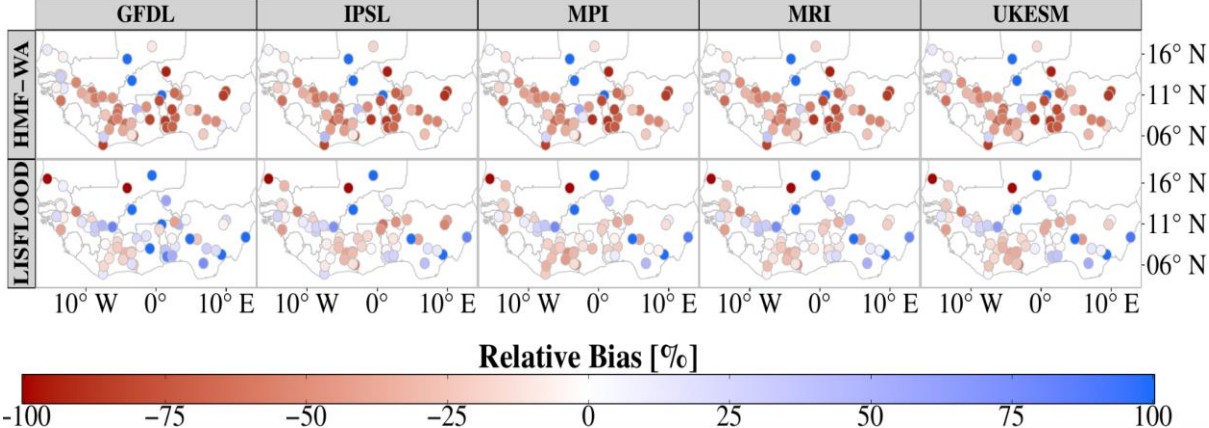

Figure 3: Relative bias (percentages) computed between simulated AMF from LISFLOOD-CMIP6 and HMFWA-CMIP6 hydrological models' simulations, and observed AMF from the ADHI database, for the historical period (1950-2014).

## 3.2 Magnitude and direction of changes in flood events

To analyse changes in floods, we have compared two 30-year future periods (a near-term future
[2031–2060] and a long-term future [2071–2100]) to a reference historical period (1985-2014).
To achieve this, we have fitted the GEV distribution the AMF series of each model simulation
using the GMLE method. Then, the 2- and 20-year flood quantiles are computed at each station
for the three 30-year periods. Figure 4 shows the MIA on the direction of changes in the 2-year
and 20-year floods for the near-term and long-term futures, from both LISFLOOD and HMF-
WA models simulations under SSP2.4-5 and SSP5.8-5 scenarios. Despite their differences in
terms of hydrological processes representation (model structures) and input data, the two
hydrological models generally projected consistent impacts of climate change on future floods
across the West African region. Both hydrological models consistently project an increase
(positive change) in floods in the near-term and long-term futures across West Africa (Figure

642    4).

In the near-term future (2031–2060), there is a high level of agreement in projecting positive
changes in the 2-year flood event under both SSP2-4.5 and SSP5-8.5 scenarios. The
simulations of the LISFLOOD and HMF-WA models show strong agreement across the
CMIP6 models. Under SSP2-4.5, the MIA values range from -0.2 to 1 for the LISFLOOD
model (Figure 4a-1), and from -0.2 to 0.8 for the HMF-WA model (Figure 4b-1). This
agreement increases for both hydrological models under SSP5-8.5, with MIA values falling
between -0.2 and 1 for both LISFLOOD (Figure 4a-3) and HMF-WA models (Figure 4b-3).
The consistent climate change impact projections suggest that more frequent flood events are
expected to become increasingly common across the West African region. For the 20-year
flood event, which is less frequent but more severe, MIA values range from -0.2 to 0.8 (-0.2 to
1) and from 0 to 0.8 (0 to 1) under the SSP2-4.5 (SSP5-8.5) for the LISFLOOD (Figure 4a-2
and Figure 4a-4) and HMF-WA (Figure 4b-2 and Figure 4b-4) models, respectively.

In the long-term future (2071–2100), considering the 2-year flood, MIA values range from -
0.6 to 1 (-0.6 to 0.8) and from -0.6 to 0.6 (0.4 to 0.8) under the SSP2-4.5 (SSP5-8.5) for the
LISFLOOD (Figure 4a-5 and Figure 4a-7) and HMF-WA (Figure 4b-5 and Figure 4b-7)
models, respectively. For the 20-year flood, model agreement in projecting the positive changes
in flood magnitude remains relatively high, with MIA values ranging from -0.4 to 0.6 (-0.4 to
0.8) and from 0 to 0.6 (-0.2 to 0.8) under the SSP2-4.5 (SSP5-8.5) for the LISFLOOD (Figure
4a-6 and Figure 4a-8) and HMF-WA (Figure 4b-6 and Figure 4b-8) models, respectively. It is
also worth noting that negative changes are projected in the 2-year flood in the long-term future

in a few sets of catchments located in the western part of the region (Figure 4a-5, 4a-7, 4b-5 and 4b-7). This area is also projected to experience a decrease in annual rainfall when looking at the full CMIP6 ensemble (IPCC, 2021). However, the agreement between the CMIP6 models remains very weak, indicating a lower confidence in the robustness of these negative changes compared to the regional pattern. Overall, the agreement between the CMIP6 and the hydrological models is higher for the near-future than for the long-term future, reflecting increased uncertainty as the projection timeline extends.

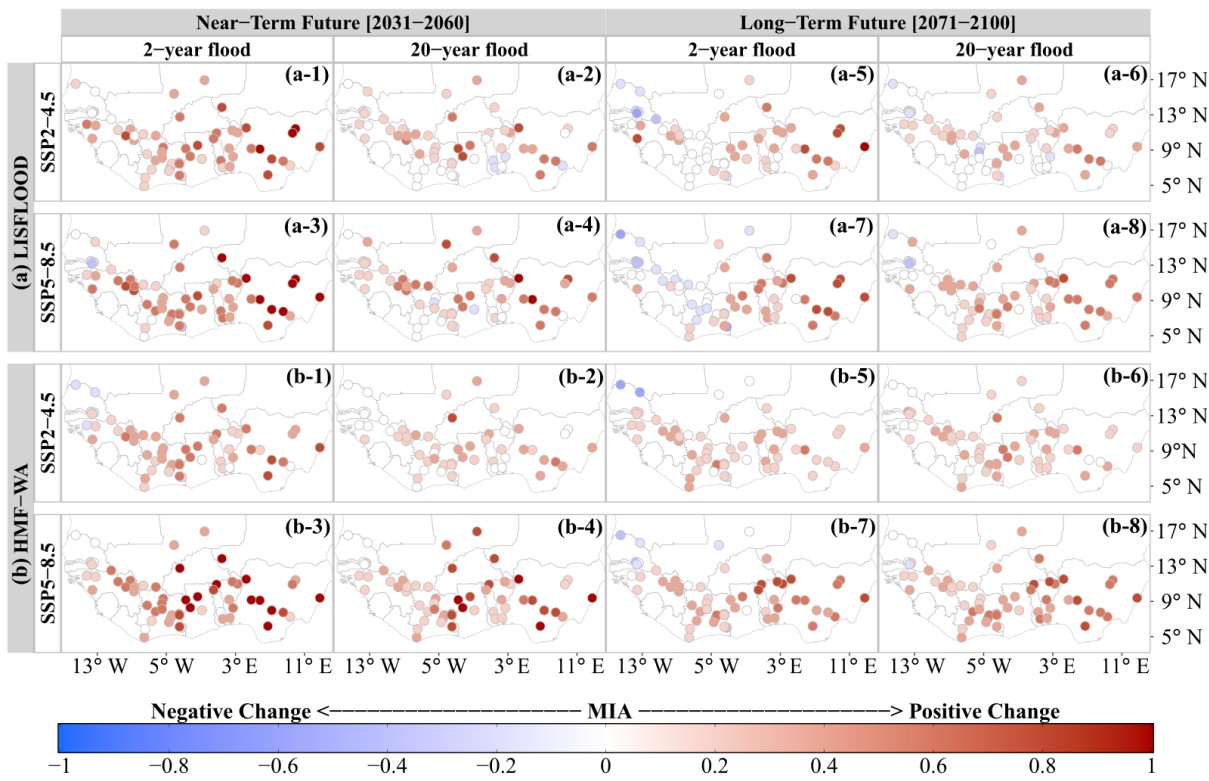

Figure 4: Spatial distribution of the multi-model index of agreement (MIA) on the direction of changes in 2-year and 20-year flood events for the near-term (2031-2060) and long-term (2071-2100) futures, compared to the historical reference period (1985-2014). This analysis combines simulations from: (a) LISFLOOD and (b) HMF-WA hydrological models, forced with five bias-corrected CMIP6 models (GFDL, IPSL, MPI, MRI, and UKESM), under the SSP2.4-5 (a1 to a4 and b1 to b4) and SSP5.8-5 (a5 to a8 and b5 to b8) scenarios. Flood quantiles are estimated using the GEV distribution fitted with the GMLE method. Negative change (decrease in flood quantiles) is represented by shades of blue, and positive change (increase in flood quantiles) is represented by shades of red.

Figure 5 summarises the projected climate impacts on floods in the near-term (2031-2060) and long-term (2071-2100) futures in West Africa across the different CMIP6 models (GFDL,

IPSL, MPI, MRI, and UKESM). Both hydrological models' simulations consistently suggest
strong changes in floods, with most median values falling above the zero-change baseline.
Considering the CMIP6 models' projections individually in the near-future, under both SSP2-
4.5 (Figure 5a) and SSP5-8.5 (Figure 5b) scenarios, the most pronounced changes are obtained
for both hydrological models when forced with IPSL, MRI, and UKESM models. These near-
term projections highlight the potential for more frequent extreme flood events, leading to
increased flood risks and greater socioeconomic vulnerability in the West African region. In
the long-term future, the distribution of flood trends is quite consistent between the two
hydrological models, and the variability stems only from GCMs. For instance, under SSP2-4.5,
the variability between the different CMIP6 models is very pronounced, with most projections
showing relatively modest changes compared to the SSP5-8.5 scenario, where most of the
GCM agree for a positive change in floods magnitudes.

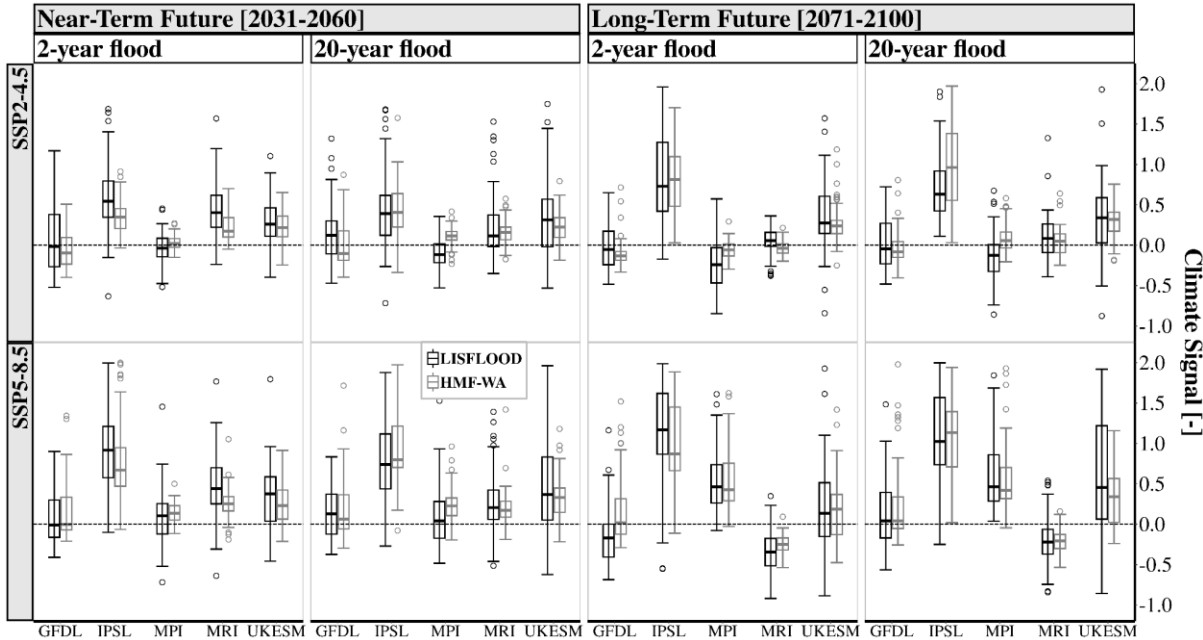


Figure 5: Synthesis of the projected changes in the 2-year and 20-year floods in West Africa
from the LISFLOOD (black boxplots) and HMF-WA (grey boxplots) model simulations forced
with the five CMIP6 GCMs (GFDL, IPSL, MPI, MRI, and UKESM), under both SSP2-4.5
(top row) and SSP5-8.5 (bottom row) climate scenarios, for the near-term (2031-2060) and the
long-term (2071-2100) futures. The climate signal (y-axis) refers to the relative change in flood
magnitude, computed as the difference between the future flood quantile (Qfuture) and the
historical flood quantile (Qhist), normalized by Qhist. The black dotted line represents the zero-
change baseline.
To further assess the agreement between the two hydrological models, Figure 7 displays how
the projected multi model mean changes in floods (ΔFlood) compares between LISFLOOD
and HMF-WA model simulations. Overall, both models project positive change in floods in
West Africa regardless of the considered SSP scenario. Indeed, most data points fall above the
zero-change baseline, indicating a global positive change in floods from both hydrological
model simulations (Figure 7). To confirm the agreement between the two models, we have
computed the Spearman coefficient (ρ) between the ΔFlood from the simulations of the
LISFLOOD and HMF-WA models.The correlation analysis shows that the agreement between
the two models is particularly pronounced under the SSP5-8.5 scenario, suggesting a stronger
influence of climatic changes under the high emissions scenario. In the near-term future, the
Spearman correlation coefficient is 0.75 (0.64) for the 2-year (20-year) floods. In the long-term
future, the correlation remains high, with 0.72 (0.70) for the 2-year (20-year) floods, suggesting
that the models continue to show strong agreement, even for long-term projections. These
results indicate a relatively high level of consistency between the two hydrological models for
projecting future flood changes, despite the systematic biases in HMF-WA model over the
reference historical period. Thus, using both models, the climate forcing has more importance
than the hydrological representation itself.

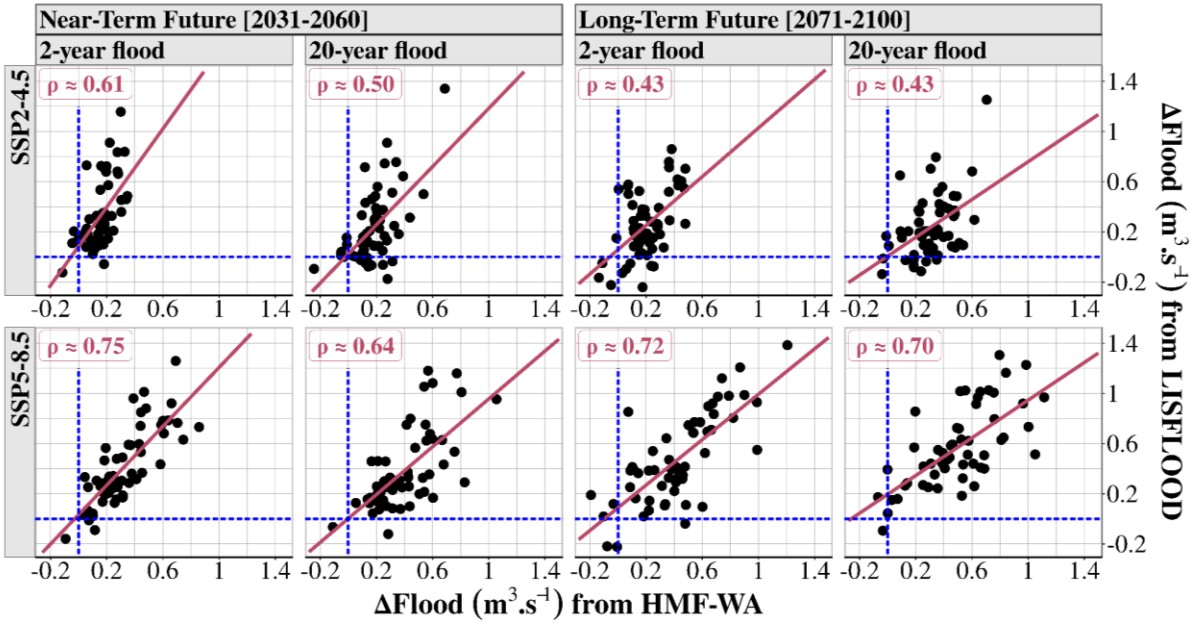

Figure 6: Comparison of projected multi model mean changes in flood (ΔFlood) between
LISFLOOD and HMF-WA hydrological models, under SSP2.4-5 (top row) and SSP5.8-5
(bottom row) scenarios, for the near-term (2031-2060) and the long-term futures (2071-2100),
compared to the historical reference period (1985-2014). The blue dashed lines represent the
zero-change baseline and the red diagonal line represents the theoretical 1:1 line where
projected changes from both hydrological models would be identical.
The relative magnitude of change in floods was also analysed by computing the mean relative
change. (i.e., ratio of the difference between the flood quantiles of the future periods and the
reference historical period) across CMIP6 models for each hydrological model. The spatial
distribution of the magnitude of changes, as simulated with the LISFLOOD and HMF-WA
hydrological models under both SSP2-4.5 and SSP5-8.5, is shown in Figure 7a and Figure 7b,
respectively. Supplementary Table S3 summarises the overall mean relative change in floods
across the region from both hydrological model's simulations. The two hydrological models
consistently project an increase in future floods across the West African region, with flood
magnitudes at most sites exceeding 50 %, particularly under SSP5-8.5 (Figure 7a-3, 7a-4, 7a-
7, 7a-8, 7b-3, 7b-4, 7b-7, and 7b-8). These results are consistent with previous studies that
argued for the ongoing rising trend in extreme streamflow across the West African catchments
(Nka et al., 2015; Aich et al., 2016; Wilcox et al., 2018; Ekolu et al., 2025). However, a
common limitation of most previous studies is their reliance on a relatively small sample of
watersheds and a limited spatial coverage, which may overlook local hydrographic variability
and limit regional applications. In addition, most impact studies in West Africa are based on
conceptual hydrological models at catchment scales. The study differs from previous studies
by covering an unprecedented set of catchments, and utilizing state-of-the-art bias-corrected
CMIP6 climate models, two large-scale hydrological models and robust statistical methods to
assess both the magnitude and field significance of future flood changes. As such, the findings
from this work provide regional-scale insights into the evolving flood risks in West Africa.
Furthermore, the findings from the studies of Almazroui et al. (2020), Dosio et al. (2021) and
Dotse et al. (2023) have shown that CMIP6 models contain a robust signal of the intensification
of the rainfall regime in West Africa. The increasing trend in floods across the region may be
partly explained by the trends in extreme precipitations, as their variability influences the
hydrological dynamics of the region (Panthou et al., 2013; Wilcox et al., 2018; Elagib et al.,

753    2021).

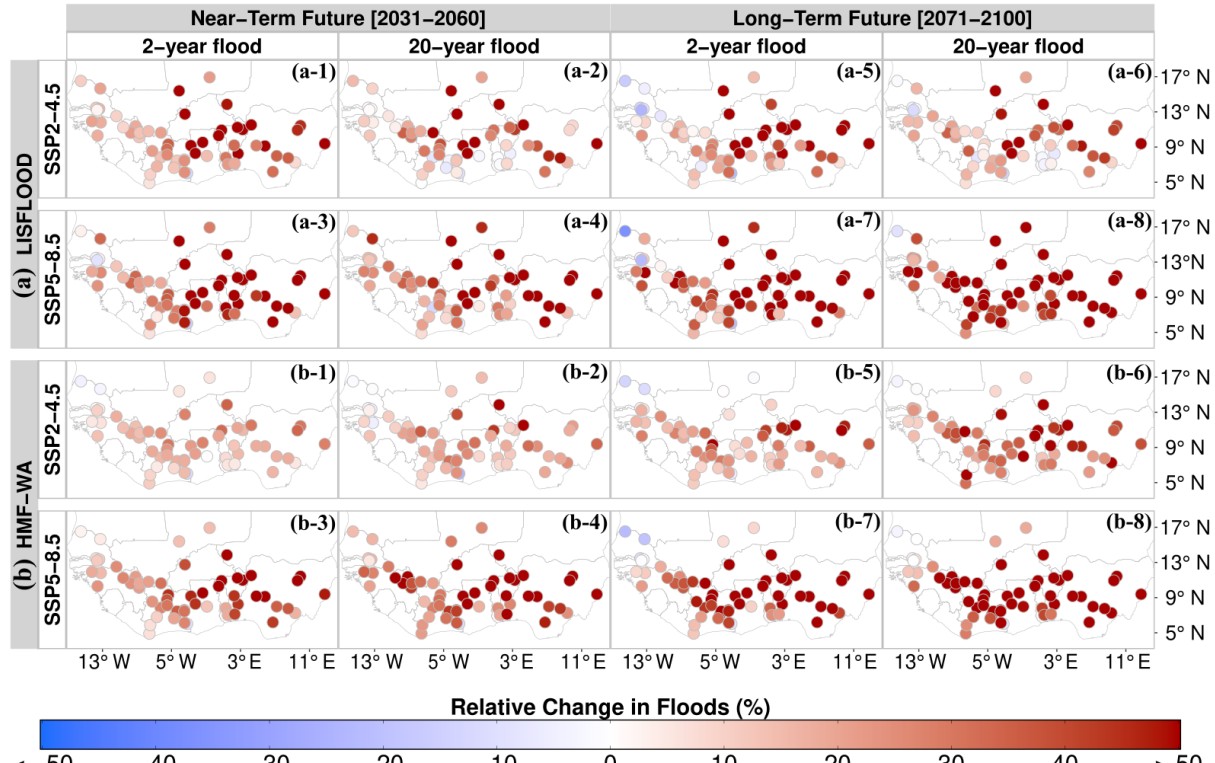

Figure 7: Mean relative changes in the 2-year and 20-year Floods in West Africa for Near-term (2031-2060) and Long-term (2071-2100) futures, based on simulations from the LISFLOOD (a-1 to a-8) and HMF-WA (b-1 to b-8) hydrological models, under SSP2-4.5 and SSP5-8.5 scenarios.

## 3.3 Onset of changes in AMF series

### 3.3.1 Observed trends in GEV Parameters

As the climate and environment change (Lee et al., 2023), it is essential to examine how these changes affect the parameters of GEV distributions. Figure 8 shows the spatial distribution of trends detected by the Mann-Kendall test on GEV parameters estimated on multi models mean AMF over 30-year moving windows from 1950 to 2100. Both hydrological models project upward trends in the location and scale parameters across the West African region with a strong agreement between the two hydrological models (see Figure 8). All local trends are field significant at 0.05 level according to the FDR procedure. The simulated upward trends in both parameters, observed across various watersheds and emission scenarios, emphasize the importance of accounting for temporal variability in GEV parameters to reliably model future flood risks. An increase in the location parameter suggests more frequent and severe floods,

while an upward trend in the scale parameter indicates greater variability in flood magnitudes. In contrast, the "mixed" trends observed in the shape parameter, with no distinct spatial patterns, support the decision to model it as constant over time, as there is no strong regional evidence of consistent temporal changes in its behaviour across the region.

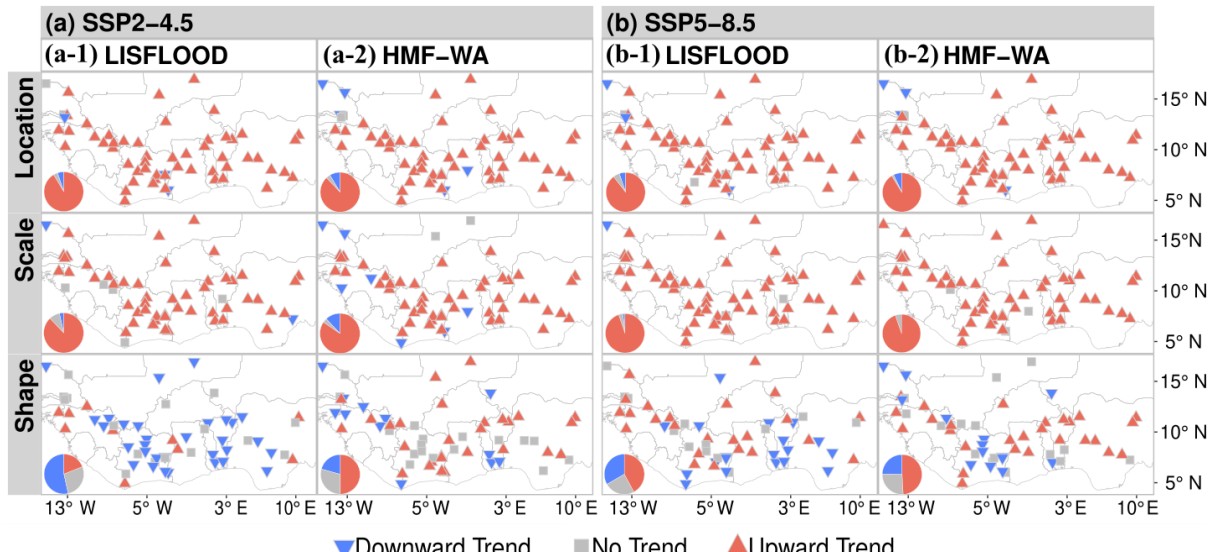

Figure 8: Direction of significant trends detected using the Mann-Kendall trend test (at the 0.05 significance level) for GEV parameters: location (top row), scale (middle row), and shape (bottom row). The GEV parameters are estimated based on multi-model mean streamflow over 30-year moving windows. Panels (a-1) and (b-1) display the results for the LISFLOOD model under SSP2-4.5 and SSP5-8.5, respectively, while panels (a-2) and (b-2) show the results for the HMF-WA model under SSP2-4.5 and SSP5-8.5, respectively. The red upward triangles indicate significant upward trends, and the blue downward triangles indicate significant downward trends, both at the 0.05 significance level. Gray rectangles represent cases where no significant trends are detected. The pie charts summarize the proportion of stations showing significant positive trends (red), significant negative trends (blue), and non-significant trends (gray).

## 3.3.2 Selection of the best-suited GEV trend model

Using non-stationary GEV models, we analyse temporal shifts in floods by fitting time-dependent GEV parameters to the AMF series from both hydrological model's simulations. To detect the onset of significant trends in flood events, we have allowed any starting year (t0) of a possible trend in the GEV location $\mu(t)$ and scale $\sigma(t)$ parameter between 1970 and 2070. To select the best non-stationary GEV model for each site, we have compared the goodness-of-fit

of three different time-varying GEV models. The models evaluated are: (1) a linear trend for
both the $\mu(t)$ and $\sigma(t)$ parameters without a breakpoint (GEV1); (2) a linear trend for $\mu(t)$ and
$\sigma(t)$ starting after a specific breakpoint (GEV2); and (3) linear trends for $\mu(t)$ and $\sigma(t)$ both
before and after a breakpoint (GEV3). Figure 9 shows the GEV trend model selected at each
station according to the AIC criterion and the deviance test for the LISFLOOD-CMIP6 and
HMFWA-CMIP6 simulations under both SSP2-4.5 and SSP-8.5 scenarios. Although both
hydrological models project an increase in floods (Figure 5), they simulate slightly different
trend patterns across the study area. Considering the LISFLOOD model (Figure 9a), the GEV3
(double linear trend) is constantly best suited at most stations, with a high agreement between
the CMIP6 models. For instance, under the SSP2-4.5 scenario, the GEV3 distribution
outperforms other models at 66 %, 79 %, 76 %, when the LISFLOOD model is driven by the
GFDL (Figure 9a-1), IPSL (Figure 9a-2) and MPI (Figure 9a-3) climate models, respectively.
A similar trend is observed under the SSP5-8.5 where the GEV3 is best suited when the
LISFLOOD is forced with the MPI (62 %), MRI (77 %), IPSL (78 %), and UKESM (66 %)
models (Figure 9a-7, 9a-8, 9a-9 and 9a-10). The HMF-WA simulations show a mixed spatial
pattern between the GEV2 and GEV3 models (Figure 9b). For both hydrological models, the
single linear trend model (GEV1) is selected at very few stations (less than 5 %). Meanwhile,
the stationary behaviour observed at few sites under SSP2-4.5 suggests that certain river basins
may experience little to no change in their hydrological extremes under moderate emissions
pathways.

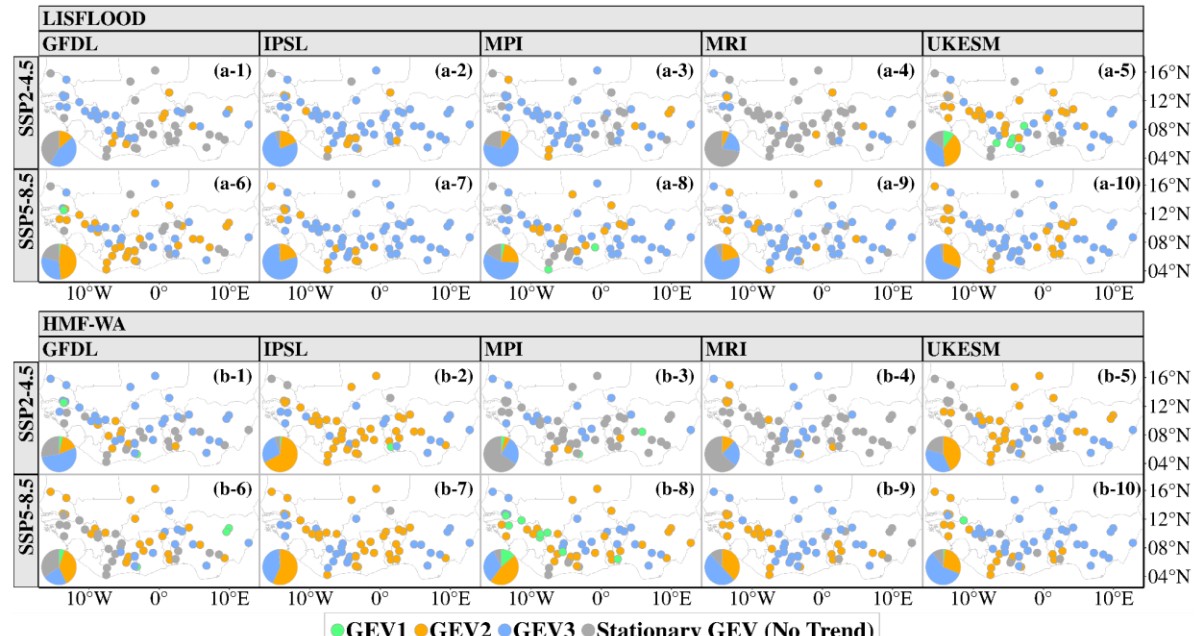


Figure 9: Best-fitting GEV trend models at each station, determined using the AIC criterion
and the deviance test, based on simulations from (a) LISFLOOD-CMIP6 (top rows) and (b)
HMF-WA-CMIP6 (bottom rows) simulations under SSP2-4.5 and SSP5-8.5 scenarios. The
green points represent stations best modelled by GEV1, which assumes a linear trend over the
entire record. The orange points indicate stations best modelled by GEV2, which assumes
stationarity before a breakpoint followed by a linear trend after the breakpoint. The blue points
denote stations best modelled by GEV3, which assumes a double linear trend. The grey points
represent stations where all non-stationary GEV models are rejected based on the deviance test.
The pie charts summarize the proportion of stations at which the stationary GEV model (grey),
or one of the non-stationary models, GEV1 (green), GEV2 (orange), or GEV3 (blue), is
identified as the best-suited for fitting the AMF series.

**3.3.3 Starting years of trends in flood hazards**
The spatial distribution of the starting years of significant flood trends detected with the GEV
trend models are shown in Figure 10. The projections from the two hydrological models are
spatially coherent, and the temporal variability on the start of flood trends in the region seems
to depend on climate models. Overall, under both SSP2-4.5 and SSP5-8.5, the majority of
significant trends are identified almost on the whole record, from the 1980s onward, in
agreement with long-term trends observed in this region (Tramblay et al., 2020), particularly
with the GFDL, IPSL, MPI, and UKESM models. This consistent pattern of early starting years
suggests that West African communities are already facing high flood risks, and are likely to

experience exacerbated conditions in the near-future. On the two linear trends in the GEV3 model, as shown in Supplementary Figure S5, the predominant spatial pattern is a transition from decreasing flood trends before the breakpoint to increasing trends after. Persistent increases, characterized by positive slopes before and after the breakpoint, are also observed at several sites, particularly with the GFDL, IPSL, and UKESM climate models.

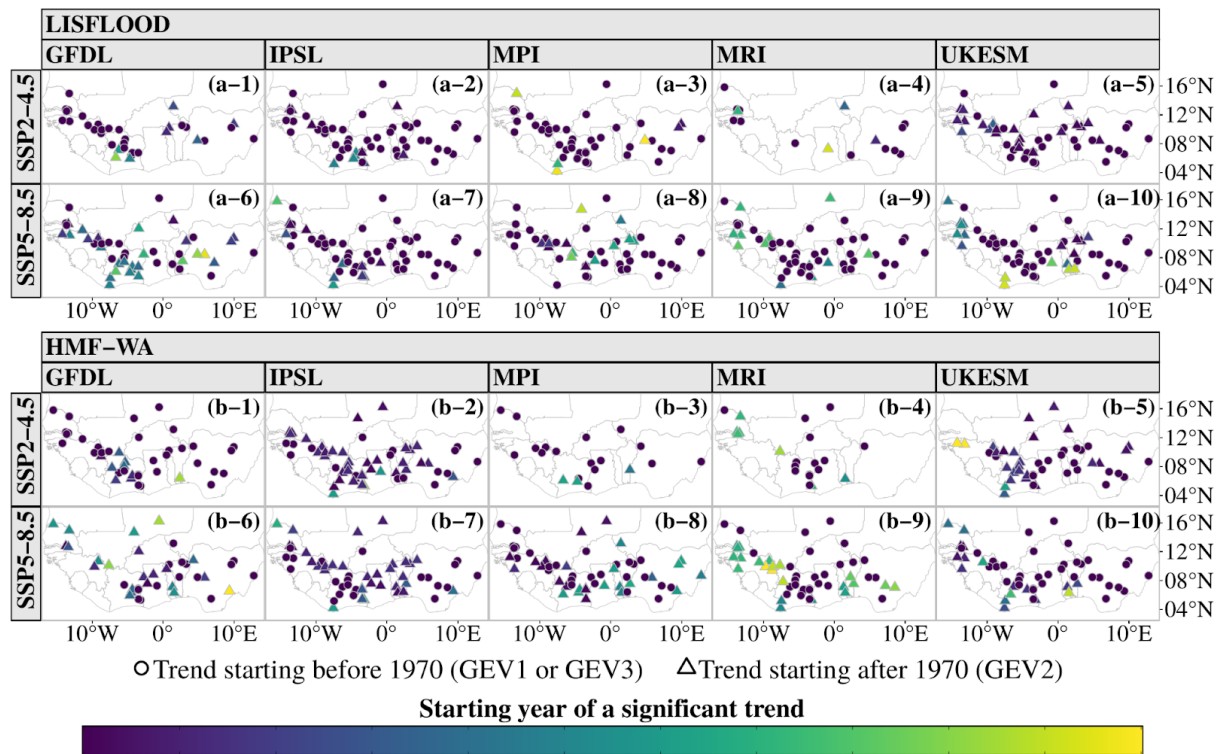

Figure 10: Spatial distribution of the starting years of significant flood trends projected by (a) LISFLOOD and (b) HMF-WA hydrological models, forced with CMIP6 models (GFDL, IPSL,MPI, MRI, and UKESM), under SSP2-4.5 and SSP5-8.5 scenarios. The color gradient indicates the starting year of a significant flood trend, ranging from 1970 (purple) to 2070 (yellow). Circular markers represent sites where trends began at the start of the time series (before 1970). Triangular markers indicate sites where trends emerged after 1970 (the linear trend GEV2 case).

## Conclusions

This study has assessed the regional-scale hydrological impacts of climate change in West Africa, specifically focusing on floods, from two large-scale hydrological models (HMF-WA and LISFLOOD) driven by five bias-corrected CMIP6 climate models under SSP2-4.5 and SSP5-8.5 scenarios. A multi-model index of agreement (MIA) was used to assess the

robustness of the projections from the hydrological model. The statistical evaluation of the two hydrological models, performed using the two-sample Anderson-Darling test between the annual maximum flows observed from the ADHI database and those simulated by the hydrological models, revealed that the LISFLOOD model outperforms the HMF-WA model in simulating extreme flows in West Africa. The GEV distribution was used to analyse trends and detect change points by fitting and comparing multiple GEV models to the AMF series, covering both the historical and future periods. Two 30-year future periods (a near-term future [2031–2060] and a long-term future [2071–2100]) were compared to a reference historical period (1985-2014). Despite differences in hydrological processes representation, model architectures and calibration, the two hydrological models generally projected consistent impacts of climate change on future floods across the West African region with a relatively high level of consistency. This agreement between the two hydrological models suggests that the climate forcing has more importance than the hydrological representation itself, and un-calibrated models can provide reliable scenarios in this region. An increase in floods (2-year and 20-year) is observed at more than 94 % of the stations, with some locations experiencing flood magnitudes exceeding 45 %. The results of the comparison between GEV trend models show that the double-linear trend GEV model with both location and scale parameters expressed as time-dependent is the best suited for most stations. The analysis of the starting years of significant flood trends revealed that most shifts in extreme flood patterns occurred early in the time series, as early as the 1970s in several basins.

The use of the GCM outputs to drive hydrological models introduces uncertainties in hydrological simulations. Indeed, the outputs of General Circulation Models (GCMs) are characterised by uncertainties, arising from several factors such as the simplified representation of complex Earth system interactions and atmospheric processes, the uncertain socioeconomic pathways, the coarse spatial resolution of these models, along with challenges related to model parameterization (Hawkins & Sutton, 2009). In addition, the performance of large-scale hydrological models is influenced by the driving inputs, the representation of the hydrological process, and the model parameterization (Andersson et al., 2015). Current models also have difficulties in reproducing hydrological processes in arid regions (Heinicke et al., 2024). It would therefore be interesting to explore in more details the main sources of uncertainties in hydrological projections in West Africa to improve the realism of such modelling approaches in the future.

## Code availability

The codes used in this study are available upon request. The implementation of these codes primarily relies on the R extRemes library (https://www.jstatsoft.org/article/view/v072i08).

## Data availability

The ADHI dataset containing the observed annual maximum time series is available at: https://doi.org/10.23708/LXGXQ9, and annual maximum dataset from the HMF-WA simulations is available at: https://doi.org/10.5285/346124fd-a0c6-490f-b5af-eaccbb26ab6b. The data that support the findings of this study are available from the corresponding author upon reasonable request.

## Author contributions

SBD, YT, and AB conceived and designed the study, with contributions from JE and BD. SBD, YT, and JB developed the methodology. YT provided the ADHI dataset and parametric bootstrapping code to assess the significance of flood trends. JE, BD, SG, and PS carried out the LISFLOOD simulations. PR provided the HMF-WA model annual maximum flow dataset. JB provided R code snippets to implement the GEV trend models. SBD performed the flood frequency analysis and drafted the initial manuscript. YT and AB supervised the study. All authors contributed to the writing and revision of the manuscript.

## Competing interest declaration

The authors declare that they have no conflict of interest.

## Acknowledgements

The PhD Grant of Serigne Bassirou Diop is funded by the AFD/IRD project CECC. The Phd Grant of Job Ekolu is funded by the Centre for Agroecology Water and Resilience (CAWR) of Coventry University, UK. The authors also extend their thanks to the various basin agencies in West Africa for their contribution to data collection and Nathalie Rouche (SIEREM) for the database management. Yves Tramblay and Bastien Dieppois were supported by a PHC ALLIANCE grant. Juliette Blanchet acknowledges receiving funding from Agence Nationale

de la Recherche - France 2030 as part of the PEPR TRACCS programme under grant number ANR-22-EXTR-0005. Ponnambalam Rameshwaran was supported by the Natural Environment Research Council as part of the NC-International program (NE/X006247/1).

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
