# Peer review of "Climate change impacts on floods in West Africa: New insight from"

_EGUsphere, 2025_

## Author Comment (AC1)

**Reviewer #1**

This manuscript presents an interesting analysis of possible future flood hazard in West Africa under climate change. Overall, the manuscript is well written and the material is well presented. It contains a few typos that I have not listed here, but I have listed a few spots below where I ask for clarification.

The manuscript builds on a range of methods that are mostly well established in the scientific literature. Thus, its methodological novelty is limited. However, I don't see this as reason for concern here. The authors have developed a meaningful workflow, integrating a rather wide range of methods. In particular, I like the systematic approach of section 2.6.3 where they determine the appropriate temporal function for the non-stationary GEV – in most papers that use the non-stationary GEV, the choice of the temporal function is based on ad-hoc decisions.

The manuscript provides important data for a region which is highly vulnerable to flooding and is characterized by data-scarcity. Hence, I would like to see this study being published.

We thank the reviewer for his overall positive evaluation of our manuscript and his constructive feedback. We are grateful that he found our study compelling and acknowledged the clarity and organization of the manuscript. Our responses to the reviewer's comments are provided below.

**Major comments:**

**Line 243: The study uses 2 SSPs, namely SSP2-4.5 and SSP5-8.5. Is this really a good choice? Please justify why you have selected (only) 2 SSPs, and particularly these 2 SSPs. Specifically, SSP5-8.5 is often criticised for being overly pessimistic.**

We thank the reviewer for this pertinent comment. We have added in the revised manuscript, at page 12, lines 367-375: "... Rather than including the full range of SSPs, we focus on SSP2-4.5 and SSP5-8.5 narratives, which represent moderate and high emission trajectories, respectively. SSP2-4.5 is considered as a "middle-of-the-road" scenario, that is consistent with current national policies and moderate progress towards emission reduction commitments (Riahi et al., 2017). In contrast, SSP5-8.5 represents a high emissions pathway, allowing us to explore the upper limits of potential impacts under continued fossil fuel dependence and minimal climate policy intervention. While SSP5-8.5 has been criticized as an "overly pessimistic" narrative (Pielke & Ritchie, 2021), it remains widely used in climate impact assessments to evaluate the vulnerability of socio-environmental systems under a "no-climate policy" world. …"

It is also important to note that the modelling work carried out here is very substantial, with 2 hydrological models applied for a large number of basins, so we have adapted the experimental plan to the available computing capacity and chosen the two most common scenarios in the literature to facilitate comparisons.

**Line 299: If I understand correctly, you use the flood time series of the region to first estimate the distribution of the GEV shape parameter via L-moments, assuming a normal distribution. Then you use this result as prior for fitting the GEV to the same flood time series. In that way, you seem to the use the same data for estimating the prior and the posterior distribution. But does this approach not violate the principle that the prior should represent information independent of the observed data? Please clarify this point.**

.

We thank the reviewer for raising this important point. It is correct that we estimate the regional distribution of the GEV shape parameter using L-moments applied to the observed AMF data, under the assumption of a normal distribution. However, we would like to clarify that this regional distribution is not used as a prior distribution to re-fit the GEV model to the same observational data. Instead, the derived distribution serves as a prior for fitting the GEV model to a separate and independent dataset, i.e., the historical and projected annual peak flood series generated by hydrological models driven by CMIP6 GCMs. This way, the prior encapsulates regional information derived from observations, but it is applied in a different modeling context. We have clarified this point in the revised manuscript (page 15, lines 353-357): "... The newly developed regional prior, modelled as a normal distribution, has a mean of -0.24 and a standard deviation of 0.16 (see Supplementary Figure S2), and is used to fit the GEV distribution to the historical and projected annual peak flood time series generated by hydrological models driven by the CMIP6 GCMs."

**Line 422: I feel that the paragraph on the field significance and FDR requires a bit more explanation to be easily followed. For example, it should be made very clear that here you look at the significance of trends when you look at all stations in your region. Also, explain what the local null hypotheses and the global null hypotheses are.**

We thank the reviewer for this valuable comment. We have updated the manuscript, page 19, lines 574-589, to provide a clearer explanation of the FDR procedure: "The null hypothesis of the deviance test assumes that the stationary GEV model provides a better fit to the data than the non-stationary model, indicating that there is no significant trend in the AMF. However, the presence of spatial cross-correlations across stations may bias the results of simultaneous multiple local tests by increasing the likelihood of detecting false positives (Farris et al., 2021). To assess the field significance of local trends detected in AMF series in the study area, we implement the False Discovery Rate (FDR) procedure (Hochberg & Benjamini, 1995). The FDR's null hypothesis assumes that none of the stations across the region exhibits a significant trend in AMF (i.e., all local null hypotheses are actually true). The FDR aims to reduce Type 1 errors (Mudge et al., 2012), by adjusting the vector of p-values from the set of at-site tests (Wilks, 2006). Due to its advantages over other methods, such as dealing with spatial autocorrelation, the FDR approach has been used in many studies of hydroclimatic variables (Khaliq et al., 2009; Chun et al., 2021). For consistency with local deviance and MK tests, the FDR procedure is computed at 0.05 global significance level ($\alpha_{global}$). The FDR test rejects the local null hypothesis when the corresponding FDR-adjusted p-value is lower than $\alpha_{global}$. Field significance is declared if the local null hypothesis is rejected at least once within the study area (Wilks, 2016)."

**Line 462: I would like to see a more detailed discussion about the causes of the difference in performance of the 2 models. The manuscript nicely lists 3 possible reasons, but it is unclear what the contribution of these reasons is. I understand that it may not be possible (within the scope of this manuscript) to estimate these contributions, but it would be helpful for the reader to better understand how important spatial resolution vs hydrological processes vs calibration is.**

We thank the reviewer for this insightful comment. We have expanded the discussion in the revised manuscript (page 21, lines 618-631) to better clarify the potential influence of each factor and the relation between these factors. Now it reads: "... Although both models are semi-physically based and spatially distributed, the LISFLOOD model outperforms the HMF-WA model in simulating extreme flows in West Africa (Figure 2). This difference in performance can be attributed to several factors: (i) the LISFLOOD model was run at a finer resolution (0.05° x 0.05°) compared to the coarser resolution of 0.1° x 0.1° used by the HMF-WA model (Rameshwaran et al., 2021); (ii) the HMF-WA model

.

includes fewer meteorological forcings and only a limited number of hydrological processes (specifically wetlands, anthropogenic water use, and endorheic rivers), whereas the LISFLOOD model can incorporate over 70 different processes depending on the target application (i.e., rainfall-runoff transformation, flood and drought forecasting) and the required level of configuration (more detailed information on the configuration of LISFLOOD can be found at https://ec-jrc.github.io/lisflood-model; and (iii) the HMF-WA model has not been calibrated to individual west African catchment conditions with observed flow data, and its performance depends on the accuracy of spatial datasets of physical and soil properties (e.g., wetlands, anthropogenic water use, and endorheic rivers) used to configure the model's hydrology to local conditions (Rameshwaran et al., 2021). In contrast, the LISFLOOD model has been regionally calibrated using in-situ discharge observations, with discharge time series spanning at least four years after 01 January 1983. Consequently, while the distributed nature of the HMF-WA model aims to improve the understanding of regional climate change impacts in a spatially coherent manner across West Africa, it does not necessarily lead to better modelling of extreme flows in the various climates and socioeconomic contexts of the region without calibration. Runoff generation is inherently a spatially distributed process. As such, the spatial resolution of a distributed hydrological model can significantly affect its ability to capture spatial variability of key watershed characteristics, such as topographic features, land cover heterogeneity, and precipitation gradients (Wolock & Price, 1994; Haddeland et al., 2002). A coarser spatial resolution limits the level of detail that can be represented in hydrological simulations, potentially overlooking important small-scale processes. Furthermore, as hydrological models are simplified representations of complex watershed processes, a calibration phase is often necessary to compensate for limited information on spatial variability of physiographical and meteorological catchments attributes, and to improve model performance in simulating the watershed's hydrological cycle (Bruneau et al., 1995). However, many river basins in West Africa have a limited number of in situ observational networks to provide the current state of hydrological information (Ndehedehe, 2019). This limits the optimal parameterization of large-scale hydrological models and may introduce uncertainties in model outputs."

**Line 613: The authors find that their "… results are consistent with previous studies that argued for the ongoing rising trend in extreme streamflow across the West African catchments ...". I assume that the current manuscript goes well beyond the papers cited that have already argued that extreme streamflow is rising. It would be good to add some explanation in what regard the current manuscript goes beyond existing studies.**

We appreciate the reviewer's insightful suggestion on adding some explanation in what regard the current manuscript goes beyond existing studies. We have added in the revised manuscript (lines 831-838): "... However, a common limitation of previous studies is their reliance on a relatively small sample of watersheds and a limited spatial coverage, which may overlook local hydrographic variability and limit regional applications. In addition, most impact studies in West Africa are based on conceptual hydrological models at catchment scales. The study differs from previous studies by covering an unprecedented set of catchments, and utilizing state-of-the-art bias-corrected CMIP6 climate models, two large-scale hydrological models and robust statistical methods to assess both the magnitude and field significance of future flood changes. As such, the findings from this work provide regional-scale insights into the evolving flood risks in West Africa."

**Line 631 and Figure 8: Why do you use here mean streamflow? Shouldn't you use AMF series?**

.

We thank the reviewer for pointing this out. The mention of mean streamflow was indeed a typographical error. The text has been corrected in the revised manuscript (line 853): "... GEV parameters estimated on multi models mean AMF …"

**Minor comments:**

**Line 91: " … updating these hydrological standards …": Please clarify what you mean by hydrological standards. Do you mean the databases?**

We thank the reviewer for this helpful remark. In this context, "hydrological standards" refers to the design flood estimations derived from existing streamflow databases. We have updated the text in the revised manuscript on page 4, lines 97-99: "... Therefore, updating these design flood estimation values (i.e. used to build dams or reservoirs) is essential to ensure that they accurately represent the current hydroclimatic context of the region (Wasko et al., 2021)."

**Line 108: Please clarify what you mean by "… the sensitivity of different climate models contrasting warming in the North Atlantic and Mediterranean Sea, which are known to influence the West African Monsoon (Bichet et al., 2020; Monerie et al., 2023), and due to contrasting emission scenarios …": Do you mean that GCMs are sensitive to the warming of both seas, and that this warming is differently simulated by different models?**

We thank the reviewer for the insightful interrogation. We mean that the uncertainties across climate models arise partly from differences in projected warming of the North Atlantic and Mediterranean Sea, explaining up to 60% of model divergences in projected rainfall changes. We have updated the text accordingly (page 4, lines 117-120): "... Nevertheless, large uncertainties remain regarding future climate trends in West Africa, partly due to differences in how climate models simulate projected warming of the North Atlantic and Mediterranean Sea, affecting the West African Monsoon and projected rainfall changes in the region (Bichet et al., 2020; IPCC, 2021; Monerie et al., 2023)…"

**Line 152: What stands the I for in "… Inter-ITCZ …"?**

The term "Inter-ITCZ" is a typo in the manuscript, and we have corrected it to "ITCZ" (Inter-Tropical Convergence Zone) in the revised version (line 172).

**Line 164: What do you mean by "… nearly half of Africa's continental watersheds are located in West Africa …"? West Africa covers about one-fifth of the African continent, but contains nearly half of the watersheds? What do you mean by continental here?**

We agree that the formulation is confusing, and have removed the word "continental" in the revised manuscript (see line 185). Now it reads: "... It is worth noting that nearly half of African watersheds are located in West Africa. …"

**Line 184: "… the white lines …": The lines are not really white.**

We have changed the word "white" by "grey" in the wording caption of Figure 1 (page 8): "Figure 1: Spatial distribution of the ADHI stations used in this study, covering the three climatic zones in the West African region, as delimited by the blue isohyets (600 mm and 1200 mm annual rainfall) on the map. The color ramp of the circles indicates the record lengths of flood data (in years). The blue lines

.

represent isohyets delimiting West African climatic regions, and the grey lines indicate the borders of West African countries."

**Line 209: These numbers are the longitude, latitude ranges? Please clarify.**

Yes, these numbers refer to the longitude and latitude ranges used in the quasi-global implementation of LISFLOOD. We have clarified this in the revised manuscript on page 10, lines 291-294: "... The LISFLOOD version used in this study (OS LISFLOOD v4.1.3) was calibrated with a 0.05° (~5 km) resolution in its quasi-global implementation covering a longitude range from -180° to 180° and a latitude range from 90° to -60° ..."

**Line 231: What is the reason for using different datasets to bias-correct the GCM output for the HMF-WA and the LISFLOOD model? I propose that you also add a bit more explanation about these 2 datasets. In which regards are they different?**

The same bias-corrected CMIP6 dataset is used to drive hydrological models, even if the bias-correction of the GCM outputs was not conducted within the scope of our study. Regarding the datasets used for bias-correction, we have added in the revised manuscript, on page 11, lines 341-357: "... The EWEMBI dataset (E2OBS, WFDEI, and ERA-I data, bias-corrected for ISIMIP; Frieler et al., 2017; Lange, 2018, 2019) was used to bias-correct the climate variables to drive the HMF-WA hydrological model. Similarly, the ERA5-land reanalysis (Muñoz-Sabater et al., 2021) was used for bias-correcting the GCMs outputs for the LISFLOOD model. The EWEMBI dataset (https://dataservices.gfz-potsdam.de/pik/showshort.php?id=escidoc:3928916) was developed to support bias correction of climate input data used in impact assessments in phase 2b of the Inter-Sectoral Impact Model Intercomparison Project (ISIMIP2b; Frieler et al., 2017). EWEMBI dataset provides global spatial coverage with 0.5° x 0.5° spatial and daily temporal resolutions. It integrates multiple sources, including ERA-Interim reanalysis data (Dee et al., 2011), the WATCH Forcing Data methodology applied to ERA-Interim (WFDEI; Weedon et al., 2014), the eartH2Observe forcing dataset (E2OBS; Calton et al., 2016), and the NASA/GEWEX Surface Radiation Budget data (SRB; (Stackhouse Jr. et al., 2011). Meanwhile, the ERA5 dataset is a global atmospheric reanalysis product developed by the Copernicus Climate Change Service (C3S) at ECMWF (European Centre for Medium-Range Weather Forecasts ReAnalysis). It is the fifth generation of atmospheric reanalysis based on 4D-Var (four-dimensional variational) data assimilation using Cycle 41r2 of the ECMWF Integrated Forecasting System (IFS) (Hersbach et al., 2020). ERA5 replaces the now outdated ERA-Interim reanalysis (Dee et al., 2011), and provides global spatial coverage from 1979 until the present, with a finer spatial and temporal resolution of 0.25° x 0.25° and 1 hour, respectively. …"

**Line 277: "… each GEV parameter […] thus guiding effective flood risk management (Lawrence, 2020)…". I was triggered by this statement and checked the associated reference (Lawrence). However, I could not find any discussion in this reference how different values of the 3 GEV parameters would guide risk management. Your following sentences explain the parameters, and you give one link between the shape parameter and design, but still I find it hard to speak of "… guiding…" in this regard. In case you want to keep this statement, then I propose that you are more explicit. For instance, would you increase the freebord of embankments for catchments/stations that have a higher scale parameter (and thus uncertainty)? Or what would you do for a catchment with a high shape parameter? Only increasing the embankments or also invest more in disaster management in contrast to a catchment with a low shape parameter?**

.

**Line 277: "… each GEV parameter […] thus guiding effective flood risk management (Lawrence, 2020)…". I was triggered by this statement and checked the associated reference (Lawrence). However, I could not find any discussion in this reference how different values of the 3 GEV parameters would guide risk management. Your following sentences explain the parameters, and you give one link between the shape parameter and design, but still I find it hard to speak of "… guiding…" in this regard. In case you want to keep this statement, then I propose that you are more explicit. For instance, would you increase the freebord of embankments for catchments/stations that have a higher scale parameter (and thus uncertainty)? Or what would you do for a catchment with a high shape parameter? Only increasing the embankments or also invest more in disaster management in contrast to a catchment with a low shape parameter?**

We thank the reviewer for this constructive observation. By the statement "guiding effective flood risk management", our intention was to convey that accurate estimation of the GEV parameters enhances the reliability of quantile estimation for design flow values, which are essential for risk-informed decision-making. However, we agree with the reviewer that Lawrence (2020) does not provide a discussion on how specific values of the GEV parameters directly inform particular risk management actions. To avoid potential misinterpretation, we have removed this statement from the revised manuscript (lines 410-411). Now the revised manuscript reads: "... In flood frequency analysis, each GEV parameter plays a distinct role in understanding and projecting flood behaviour (Lawrence, 2020; Wasko et al, 2021) ..."

**Line 315: "… stations at which the null hypothesis … is rejected …": Please specify what exactly the null hypothesis is. Is it that the simulated annual flood peaks follow the same distribution as the observed flood peaks? If yes, does this mean that you use all stations, independent of whether the models represent the observed floods well?**

The null hypothesis in the AD test assumes that the simulated and observed annual flood peaks follow the same statistical distribution. Thus, we are testing whether the hydrological models are able to reproduce the statistical behavior of observed annual maximum flows at each station. We have added in the methodological section (section 2.5, lines 383-384): "...The null hypothesis of the AD test assumes that the simulated and observed AMF follow the same statistical distribution ...". Regarding the use of all stations, the null hypothesis is more frequently rejected for the non-calibrated HMF-WA model. As noted in the original manuscript, "whether a calibrated hydrological model offers more reliable climate change projections than an uncalibrated model, which may perform less accurately in reproducing historical conditions (Pechlivanidis et al., 2017), remains questionable". Moreover, one of the objectives of our study is to assess the regional consistency of projections from two structurally different hydrological models: one calibrated, the other not. As such, we have decided to keep all models and stations to analyze the difference in the projections.

**Line 362: Here SGEV occurs, but this abbreviation is only explained later.**

We thank the reviewer for pointing this out. We have introduced the abbreviation "SGEV" earlier in the text, at its first occurrence on line 513: "... to fit the stationary GEV model (SGEV) … "

**Line 442: What exactly do you mean by "… the hydrological model is considered to perform poorly at that station…"? Have you discarded these models? If yes, what does this mean for the following analyses and metrics? For instance, then you probably do not have 5 models at each station in Figure 2.**

.

By stating that "... the hydrological model is considered to perform poorly at that station ...", we refer to cases where the hydrological model shows a statistically significant mismatch between the simulated and observed annual maximum flows. We would like to clarify that the AD test is only used to assess regional-scale performance of hydrological models, and not as a filtering criterion for inclusion or exclusion of models or stations. For this reason, all model combinations (i.e., hydrological model and GCM pairings) are used in the analysis. We have added the following statement in the revised manuscript (lines 395-397): "... It is important to note that the AD test is only used herein to assess regional-scale performance of hydrological models, and not as a filtering criterion for inclusion or exclusion of models or stations."

**Figure 2: I find the markers a bit disturbing, and the color not easy to see. Wouldn't it be easier to have filled markers (circles and real squares – not these squares divided into 4 smaller squares)?**

We thank the reviewer for his constructive suggestion. We have updated the Figure 2 accordingly (see below and page 23 in revised manuscript).

**Line 472: Circles and squares show 60-100% and 0-20%, respectively. How about stations with 20-40% failure rate? And you do you use these 2 classes? Why not using 6 classes (from 5 out of 5 models fail to 0 out of 5 models fail)?**

We have updated the Figure 2 to use 6 classes (from 5 out of 5 models fail to 0 out of 5 models fail, as suggested by the reviewer (see page 23 in revised manuscript):

.

[Figure]

Figure 2: Statistical evaluation of the two hydrological models: a) Two-sample Anderson-Darling (AD) goodness-of-fit (GOF) test at 0.05 statistical significance level at each station between the AMF of daily OBS from the ADHI database and annual maxima flow of HIST from LISFLOOD daily simulations forced with the five CMIP6 GCMs (GFDL, IPSL, MPI, MRI, and UKESM) over the period 1950-2014. b) same as a) but using HMF-WA as hydrological model. The fill color of the markers indicates the proportion of CMIP6 models (out of five) for which the AD test null hypothesis (i.e., simulated and observed AMF follow the same statistical distribution) is rejected at the 0.05 significance level. Marker shapes correspond to binned categories of this proportion, as indicated in the legend.

**Figure 5: Please explain in the figure caption what 'climate signal' is – I assume it is the ratio that you have introduced in Line 324, but this should be clear.**

We thank the reviewer for this helpful suggestion. We have updated the caption of Figure 5 to clarify the term "climate signal" in the revised manuscript, at page 28: "... The climate signal refers to the relative change in flood magnitude, computed as the difference between the future flood quantile (Qfuture) and the historical flood quantile (Qhist), normalized by Qhist. …"

.

[Figure]

Figure 5: Synthesis of the projected changes in the 2-year and 20-year floods in West Africa from the LISFLOOD (black boxplots) and HMF-WA (grey boxplots) model simulations forced with the five CMIP6 GCMs (GFDL, IPSL, MPI, MRI, and UKESM), under both SSP2-4.5 (top row) and SSP5-8.5 (bottom row) climate scenarios, for the near-term (2031-2060) and the long-term (2071-2100) futures. The climate signal represents the relative change in flood magnitude compared to the historical baseline (1985–2014), The climate signal refers to the relative change in flood magnitude, computed as the difference between the future flood quantile (Qfuture) and the historical flood quantile (Qhist), normalized by Qhist. The black dotted line denotes the baseline (i.e., no change).

**Line 579: Delta Flood is mentioned here, but the exact definition follows only a few lines later.**

We have revised the paragraph, on page 28, lines 783-790, to introduce the definition of Delta Flood (ΔFlood) upon its first mention: "To further assess the agreement between the two hydrological models, Figure 7 displays how the projected multi model mean changes in floods (ΔFlood) compares between LISFLOOD and HMF-WA model simulations. Overall, both models project positive change in floods in West Africa regardless of the considered SSP scenario. Indeed, most data points fall above the zero-change baseline, indicating a global positive change in floods from both hydrological model simulations (Figure 7). To confirm the agreement between the two models, we have computed the Spearman coefficient (ρ) between the ΔFlood from the simulations of the LISFLOOD and HMF-WA models …"

**Figure 6: You could add the correlation coefficients in the sub-plots. That would substitute Table S1, and the reader would have the summary metrics directly when he/she looks at Figure 6.**

We appreciate this helpful suggestion. We have removed the Table S1 in supplementary material, updated Figure 6, and now added the Spearman correlation coefficients (ρ) directly to each subplot (see page 30 in the revised manuscript):

.

[Figure]

Figure 6: Comparison of projected multi model mean changes in flood (ΔFlood) between LISFLOOD and HMF-WA hydrological models, under SSP2.4-5 (top row) and SSP5.8-5 (bottom row) scenarios, for the near-term (2031-2060) and the long-term futures (2071-2100), compared to the historical reference period (1985-2014). The blue dashed lines represent the zero-change baseline and the red diagonal line represents the theoretical 1:1 line where projected changes from both hydrological models would be identical.

**Figure 9: You could add pie charts in the sub-plots as in Figure 8. That would summarize the information for the reader.**

We thank the reviewer for the helpful suggestion We have now added pie charts in each sub-plot of Figure 9 (see page 35 in the revised manuscript):

[Figure]

Figure 9: Best-fitting GEV trend models at each station, determined using the AIC criterion and the deviance test, based on simulations from (a) LISFLOOD-CMIP6 (top rows) and (b) HMF-WA-CMIP6

.

(bottom rows) simulations under SSP2-4.5 and SSP5-8.5 scenarios. The green points represent stations best modelled by GEV1, which assumes a linear trend over the entire record. The orange points indicate stations best modelled by GEV2, which assumes stationarity before a breakpoint followed by a linear trend after the breakpoint. The blue points denote stations best modelled by GEV3, which assumes a double linear trend. The grey points represent stations where all non-stationary GEV models are rejected based on the deviance test. The pie charts summarize the proportion of stations at which the stationary GEV model (grey), or one of the non-stationary models, GEV1 (green), GEV2 (orange), or GEV3 (blue), is identified as the best-suited for fitting the AMF series.

**Figure 10: (1) The GEV1 model shows a linear trend over the entire time period, correct? Then its starting year will be in the first year of the time period. Does it really make sense to say there is a significant breakpoint? A linear trend has no breakpoint. (2) The GEV3 model has 2 linear trends with a breakpoint in-between. Wouldn't it be more interesting to show the breakpoint in-between? And I think it would be interesting to understand how these 2 trends look like, for example, do we have cases where floods decrease before the breakpoint and increase after it? (3) The starting years / colors are not so easy to see. Maybe use filled markers.**

For points (1) and (3), as stated in the methodological section 2.6.3, the GEV1 model represents "a linear trend with no breakpoint". Thus, it does not make sense to refer to a "significant breakpoint" in this context since the starting year of significant change in a linear trend will be the first year of the time period. We have updated the wording in the caption of Figure 10 accordingly (see page 37 in the revised manuscript). Regarding point (3), we have updated Figure 10 using filled markers to improve visual clarity in the revised manuscript (page 37):

[Figure]

Figure 10: Spatial distribution of the starting years of significant flood trends projected by (a) LISFLOOD and (b) HMF-WA hydrological models, forced with CMIP6 models (GFDL, IPSL, MPI, MRI, and UKESM), under SSP2-4.5 and SSP5-8.5 scenarios. The color gradient indicates the starting year of a significant flood trend, ranging from 1970 (purple) to 2070 (yellow). Circular markers

.

represent sites where trends began at the start of the time series (before 1970). Triangular markers indicate sites where trends emerged after 1970 (the linear trend GEV2 case).

Regarding point (2), we have added a short paragraph in the revised manuscript, at page 35, lines 936-940, to describe how the two trends in the GEV3 model behave before and after the breakpoint: "... On the two linear trends in the GEV3 model, as shown in Supplementary Figure S4, the predominant spatial pattern is a transition from decreasing flood trends before the breakpoint to increasing trends after. Persistent increases, characterized by positive slopes before and after the breakpoint, are also observed at several sites, particularly with the GFDL, IPSL, and UKESM climate models. ...". This paragraph is further supported by Supplementary Figure S5, which shows the spatial distribution of the trend directions derived from the non-stationary GEV3 model:

[Figure]

Figure S5: Spatial distribution of flood trend direction derived from the non-stationary GEV3 model, based on annual maximum flood (AMF) series simulated by two hydrological models (LISFLOOD and HMF-WA), forced with five CMIP6 GCMs (GFDL, IPSL, MPI, MRI, UKESM) under SSP2-4.5 and SSP5-8.5 scenarios. The green downward triangles represent decreasing trends before and after the breakpoint, the orange circles indicate decreasing trend before and increasing trend after, the blue regular rectangles show increasing trend before and decreasing trend after, and the red upward rectangles correspond to increasing trends before and after the breakpoint.


[Figure]

Figure S2: Illustration showing the handling of missing data in an annual hydrograph of daily discharge measurements. A significant portion of data, particularly around the peak discharge period, is missing (highlighted by the red circle). Such a year is excluded from the analysis to ensure the accuracy of the annual peak flood sampling.

.

**- How variable is the land uses in this study region? If they are rather heterogeneous, a summary of key land use types in each catchment would also be useful.**

The study area is characterized by a heterogeneous landscape across the different catchments, with considerable variability in land use. We have added to the Supplementary Material (Table S2) a summary of key land cover types (forest, urban, crop, irrigated crops, grass, shrub, sparse, and bare) in each catchment, detailing the dominant land uses and their respective proportions.

Table S2: Land use distribution across different catchments in the study area. The table shows the proportion of each land use type (Forest, Urban, Crop, Irrigated Crops, Grass, Shrub, Sparse, and Bare) within the catchments identified by their unique IDs. Each value represents the proportion (percentage) of the respective land use type within a given catchment.

| ID | Forest | Urban | Crop | Crop Irrig | Grass | Shrub | Sparse | Bare |
|---|---|---|---|---|---|---|---|---|
| ADHI_114 | 0.06 | 0 | 0.46 | 0 | 0 | 0.48 | 0 | 0 |
| ADHI_121 | 0.27 | 0 | 0.29 | 0 | 0 | 0.44 | 0 | 0 |
| ADHI_123 | 0.83 | 0 | 0.11 | 0 | 0 | 0.06 | 0 | 0 |
| ADHI_131 | 0.74 | 0 | 0.17 | 0 | 0 | 0.09 | 0 | 0 |
| ADHI_144 | 0.49 | 0 | 0.49 | 0 | 0 | 0.02 | 0 | 0 |
| ADHI_163 | 0.96 | 0 | 0.01 | 0 | 0 | 0.03 | 0 | 0 |
| ADHI_172 | 0.05 | 0 | 0.78 | 0.01 | 0.02 | 0.14 | 0 | 0 |
| ADHI_179 | 0.27 | 0.01 | 0.72 | 0 | 0 | 0 | 0 | 0 |
| ADHI_180 | 0.31 | 0 | 0.67 | 0 | 0 | 0 | 0 | 0 |
| ADHI_183 | 0.56 | 0 | 0.44 | 0 | 0 | 0 | 0 | 0 |
| ADHI_187 | 0.33 | 0 | 0.39 | 0 | 0 | 0.27 | 0 | 0 |
| ADHI_198 | 0.52 | 0 | 0.25 | 0 | 0 | 0.23 | 0 | 0 |
| ADHI_270 | 0.76 | 0 | 0.17 | 0 | 0 | 0.07 | 0 | 0 |
| ADHI_276 | 0.74 | 0 | 0.14 | 0 | 0 | 0.12 | 0 | 0 |
| ADHI_304 | 0.02 | 0 | 0.04 | 0.03 | 0.25 | 0.37 | 0.26 | 0.07 |
| ADHI_315 | 0.05 | 0 | 0.74 | 0 | 0 | 0.2 | 0 | 0 |
| ADHI_316 | 0.09 | 0 | 0.41 | 0.01 | 0.11 | 0.08 | 0.05 | 0.25 |
| ADHI_319 | 0.14 | 0 | 0.38 | 0.01 | 0.09 | 0.11 | 0.03 | 0.25 |
| ADHI_320 | 0 | 0 | 0.62 | 0.02 | 0.21 | 0 | 0.12 | 0.03 |
| ADHI_321 | 0.33 | 0 | 0.51 | 0 | 0.01 | 0.13 | 0 | 0 |
| ADHI_324 | 0.37 | 0 | 0.48 | 0.01 | 0 | 0.14 | 0 | 0 |
| ADHI_325 | 0.83 | 0 | 0.09 | 0 | 0.04 | 0.03 | 0 | 0 |
| ADHI_332 | 0.39 | 0 | 0.02 | 0 | 0.58 | 0.01 | 0 | 0 |
| ADHI_372 | 0.02 | 0 | 0.8 | 0 | 0 | 0.17 | 0 | 0 |
| ADHI_390 | 0.7 | 0 | 0.23 | 0 | 0 | 0.07 | 0 | 0 |
| ADHI_394 | 0.62 | 0 | 0.33 | 0 | 0 | 0.04 | 0 | 0 |
| ADHI_507 | 0.58 | 0 | 0.18 | 0 | 0 | 0.24 | 0 | 0 |
| ADHI_510 | 0.8 | 0 | 0.08 | 0 | 0 | 0.11 | 0 | 0 |
| ADHI_511 | 0.43 | 0 | 0.44 | 0 | 0 | 0.13 | 0 | 0 |
| ADHI_515 | 0.32 | 0 | 0.54 | 0 | 0 | 0.14 | 0 | 0 |
| ADHI_519 | 0.05 | 0 | 0.74 | 0 | 0.09 | 0.11 | 0 | 0 |
| ADHI_531 | 0.5 | 0 | 0.44 | 0 | 0 | 0.05 | 0 | 0 |
| ADHI_548 | 0.78 | 0 | 0.01 | 0 | 0 | 0.2 | 0 | 0 |
| ADHI_550 | 0 | 0 | 0.21 | 0.01 | 0.69 | 0.01 | 0.05 | 0.03 |
| ADHI_560 | 0.08 | 0 | 0.38 | 0.02 | 0.23 | 0.2 | 0.04 | 0.06 |
| ADHI_571 | 0.11 | 0 | 0.49 | 0 | 0 | 0.39 | 0 | 0 |
| ADHI_585 | 0.63 | 0 | 0.29 | 0 | 0 | 0.07 | 0 | 0 |
| ADHI_587 | 0.56 | 0 | 0.23 | 0 | 0 | 0.21 | 0 | 0 |
| ADHI_592 | 0.1 | 0.01 | 0.87 | 0 | 0 | 0.02 | 0 | 0 |

.

| | | | | | | | | |
|---|---|---|---|---|---|---|---|---|
| ADHI_595 | 0.19 | 0 | 0.58 | 0.01 | 0.01 | 0.21 | 0 | 0 |
| ADHI_596 | 0.75 | 0 | 0.15 | 0 | 0 | 0.09 | 0 | 0 |
| ADHI_597 | 0.96 | 0 | 0.01 | 0 | 0 | 0.03 | 0 | 0 |
| ADHI_605 | 0.02 | 0 | 0.93 | 0.01 | 0 | 0.05 | 0 | 0 |
| ADHI_607 | 0.43 | 0 | 0.09 | 0 | 0 | 0.48 | 0 | 0 |
| ADHI_612 | 0.39 | 0 | 0.09 | 0 | 0 | 0.52 | 0 | 0 |
| ADHI_613 | 0.71 | 0 | 0.07 | 0 | 0 | 0.21 | 0 | 0 |
| ADHI_617 | 0.61 | 0 | 0.21 | 0 | 0 | 0.18 | 0 | 0 |
| ADHI_639 | 0.41 | 0 | 0.55 | 0 | 0 | 0.04 | 0 | 0 |
| ADHI_640 | 0.3 | 0 | 0.57 | 0.01 | 0.01 | 0.11 | 0 | 0 |
| ADHI_649 | 0.91 | 0 | 0.07 | 0 | 0 | 0.02 | 0 | 0 |
| ADHI_650 | 0.92 | 0 | 0.01 | 0 | 0 | 0.07 | 0 | 0 |
| ADHI_651 | 0.72 | 0 | 0.16 | 0 | 0 | 0.11 | 0 | 0 |
| ADHI_678 | 0.76 | 0 | 0.1 | 0 | 0 | 0.14 | 0 | 0 |
| ADHI_692 | 0.1 | 0 | 0.45 | 0.03 | 0.19 | 0.12 | 0.04 | 0.08 |
| ADHI_1183 | 0.19 | 0 | 0.4 | 0 | 0 | 0.4 | 0 | 0 |
| ADHI_1269 | 0.71 | 0.01 | 0.28 | 0 | 0 | 0.01 | 0 | 0 |
| ADHI_1400 | 0.07 | 0 | 0.8 | 0 | 0 | 0.13 | 0 | 0 |
| ADHI_1401 | 0.1 | 0 | 0.75 | 0 | 0 | 0.15 | 0 | 0 |

**- Sources of input data for the hydrological models e.g., rainfall, temperature – it is unclear where they are from, it is implied from the later Section 2.5 that rainfall and temperature were from GCM rather than observed, but it would be helpful to clarify this earlier in the data section.**

We appreciate the helpful suggestion from the reviewer. In response, we have revised Section 2.2 (lines 201–226) to clarify the sources of input data for the hydrological models. Now Section 2.2 reads: "**2.2 Observational data and climate forcings for hydrological experiments:** Daily streamflow data for the period 1950-2018 were obtained from the African Database of Hydrometric Indices (ADHI) recently developed by Tramblay et al. (2021). This database provides hydrometric indices computed from different data sources, with daily discharge time series that span at least 10 years. In the ADHI database, the size of the 441 West African catchments ranges from 95 to 2,150,000 km$^2$, and some stations have daily discharge data spanning over 44 years. Figure 1 shows the spatial distribution of the ADHI stations used in this study. We only selected watersheds from the ADHI database that met the following three criteria: (i) low regulation, determined through visual inspection of dam locations relative to watershed outlets (see Supplementary Figure S1), combined with a year-by-year analysis of annual hydrographs to assess the impact of dam operations on streamflow, (ii) surface area of less than 150,000 km², and (iii) a daily streamflow time series covering a minimum of 10 years between the 1950 and 2018. To address the challenges associated with missing data in the database, we conducted a visual inspection of hydrographs at each station as illustrated by Supplementary Figure S2. Years with data gaps near the flood peak were excluded from the analysis to avoid the risk of missing the true annual peak flood (Wilcox et al., 2018). Through this careful screening process, we ensured that no AMF values were derived from periods characterized by a lot of missing data. It is important to note that the observational streamflow data are not used to calibrate or drive the hydrological models. Instead, these observations serve as an independent benchmark to evaluate the ability of the hydrological models to reproduce key flood statistics during the historical period. The LISFLOOD model was calibrated using the ERA5 reanalysis dataset, which provides consistent and high-resolution precipitation and temperature fields. Moreover, ERA5 was also used as a reference for the bias correction of the five climate models from the CMIP6 ensemble that were used to drive the hydrological simulations for both the historical and future periods (see Section 2.4)."

.

**4. Section 2.3: I understand that the details of the two hydrological models are presented in the corresponding papers cited, but I think the readers could benefit from some additional background on these models, at least covering the key processes represented in each model on converting rainfall to runoff. This information is currently only partly available for the HWF-WA model (with only the recently added process representations listed) and not communicated for the LISFLOOD model. After presenting these, I'd also love to see a quick summary of the key differences between the models to justify your point in the Abstract that the two models 'differ in their hydrological process representation'.**

We appreciate the reviewer's suggestion. We have expanded Section 2.3 (lines 261-319) to clarify the key hydrological processes in each model and the key differences between the two models. The updated section now reads: "The HMF-WA model is adapted from the modular HMF model, and is designed for large-scale applications across West Africa (Rameshwaran et al., 2021). It employs a vertically integrated soil moisture scheme to simulate runoff production, driven by rainfall and potential evaporation inputs. Runoff generation considers soil drainage and a spatial probability distribution of soil moisture. Routing is based on a kinematic wave approach (Bell et al., 2007), with parallel pathways for surface and subsurface flow. Key enhancements over the classical HMF model include modules to simulate wetland inundation, endorheic basins, and anthropogenic water withdrawals, making it well-suited for semi-arid environments with complex hydrology (Rameshwaran et al., 2021). HMF-WA simulates spatially consistent river flows across West Africa at a 0.1° × 0.1° spatial resolution. Although it has not yet been specifically calibrated to individual West African catchments using observed streamflow data where the model hydrology is configured to local conditions using spatial datasets of physical and soil properties, HMF-WA model evaluation against observational data indicates that it performs reasonably well in simulating both daily high and low river flows across most catchments. The median values of NSE (Nash-Sutcliffe efficiency), NSElog, and BIAS are 0.62, 0.82, and 0.06 (6 %), respectively (Rameshwaran et al., 2021).

The LISFLOOD model, developed by the Joint Research Centre (JRC) of the European Commission (https://ec-jrc.github.io/lisflood/), is a physical, spatially distributed hydrological model, designed for simulating several hydrological processes that occur in a catchment (Van Der Knijff et al., 2010). The LISFLOOD model simulates water processes using a three-layer soil water balance, along with groundwater and subsurface flow models. It accounts for several processes such as snow accumulation/melt, infiltration, evapotranspiration, groundwater flow, surface runoff, etc. Moreover, it supports the integration of human influences such as reservoirs and water abstraction. The numerical LISFLOOD simulation is driven by meteorological forcing (precipitation, temperature, and evapotranspiration) combined with high-resolution spatial data on terrain morphology, soil characteristics, land use, and water demand. This integrated setup allows the model to simulate runoff processes under diverse climatic and socio-economic conditions, capturing both natural and anthropogenic influences across heterogeneous landscapes. The runoff produced at every grid cell within the model domain is routed through the river network using a kinematic wave approach. The LISFLOOD version used in this study (OS LISFLOOD v4.1.3) was regionally calibrated with a 0.05° (~5 km) resolution, using in-situ discharge gauge stations with at least four years of daily measurements recorded after 1 January 1982. In this setup, model parameters are linked to global geospatial datasets describing catchment morphology and river networks, land use, vegetation characteristics, soil properties, lake distribution, and water demand (Salamon et al., 2024; Choulga et al., 2024). The Distributed Evolutionary Algorithms in Python (DEAP; Fortin et al., 2012) framework was applied to optimize parameters in gauged catchments, with the modified Kling-Gupta Efficiency (KGE; Gupta et al., 2009) utilized as the objective function. Calibration was performed over a continuous simulation

.

period using ERA5 reanalysis meteorological forcing. Due to the varying length and temporal coverage of the discharge records used for calibration, model performance was assessed using all available observational data at each station, rather than splitting the records into separate calibration and validation periods. The LISFLOOD calibration tool is freely available at https://github.com/ec-jrc/lisflood-calibration.

Globally, while both models use a kinematic wave routing scheme, HMF-WA and LISFLOOD differ significantly in their hydrological process representation. HMF-WA applies a vertically integrated soil moisture scheme with simplified runoff generation based on spatial soil moisture distribution. In contrast, LISFLOOD features a more detailed, physically-based three-layer soil model with an explicit representation of groundwater, snow processes, and anthropogenic influences. Furthermore, LISFLOOD has been calibrated using in-situ discharge data. Nevertheless, while calibration can enhance the accuracy of discharge simulations, several studies have highlighted that uncalibrated global hydrological models often exhibit comparable sensitivity to climate variability as the regional calibrated hydrological models, particularly when assessing relative changes in extreme events between future and historical periods (Gosling et al., 2017; Zhao et al., 2025). Therefore, whether a calibrated hydrological model offers different climate change projections than an uncalibrated model needs further investigation (Pechlivanidis et al., 2017).”

**Specific comments:**

**1. Line 51 – it will be clearer if the change in flood magnitude can be summarized specific to the flood return period(s) investigated.**

We appreciate the reviewer's suggestion. We now specify the change in flood magnitude by return period and future horizon in the abstract (lines 52-55): “... Flood magnitudes are projected to increase at 94% (96%) of stations for the 2-year (20-year) event in the near-term future, and at 88% (93%) of stations for the 2-year (20-year) event in the long-term future, with some locations expected to experience increases exceeding 45%. …”

**2. Figure 1 caption: ‘grey lines’ instead of ‘white lines’?**

We have changed the word “white” by “grey” in the wording caption of Figure 1 (page 8): “Figure 1: Spatial distribution of the ADHI stations used in this study, covering the three climatic zones in the West African region, as delimited by the blue isohyets (600 mm and 1200 mm annual rainfall) on the map. The color ramp of the circles indicates the record lengths of flood data (in years). The blue lines represent isohyets delimiting West African climatic regions, and the grey lines indicate the borders of West African countries.”

**3. Line 177 – decision on ‘low regulation’ catchments: the Supplementary Fig. 1 suggested that this is based on whether there is a dam located near the watershed outlet, with no information what defines a dam ‘near’ or ‘far from’ the outlet – was this based on visual inspection, or a threshold distance used? If the latter, how was the threshold distance determined?**

We appreciate the reviewer’s constructive comment regarding the decision on ‘low regulation’ catchments. The identification of "low regulation" catchments was based on visual inspection of the dam locations relative to the watershed outlet, using both the GRanD database (https://www.globaldamwatch.org/grand) and Google maps, combined with a year-by-year analysis of the annual hydrographs. This allowed us to verify whether the dam's construction or operational start

.

date caused noticeable changes in the streamflow regime. No fixed distance threshold was applied. We have clarified this aspect in the description of the observational data, Section 2.2, lines 208-213: "... We only selected watersheds from the ADHI database that met the following three criteria: (i) low regulation, determined through visual inspection of dam locations relative to watershed outlets (see Supplementary Figure S1), combined with a year-by-year analysis of annual hydrographs to assess the impact of dam operations on streamflow, (ii) surface area of less than 150,000 km², and (iii) a daily streamflow time series covering a minimum of 10 years between the 1950 and 2018. …"

**4. Section 2.6.1 – the introductory section for GEV is very informative, however, I think it could benefit from additional information on what positive and negative shape parameters mean, which seem to be useful context to the subsequent discussion on the plausible values of the shape parameter.**

We thank the reviewer for the insightful suggestion. We have now added information about the meaning of shape parameter values, in Section 2.6.1, lines 418-426: "... The shape parameter ($\xi$) governs the tail behaviour of the GEV distribution, which encompasses three types of extreme value distributions (Coles, 2001): (i) a positive shape parameter ($\xi > 0$) indicates a heavy-tailed Fréchet case (Fréchet, 1927), suggesting an increased probability of extreme flooding events, (ii) a null shape parameter ($\xi = 0$) suggests a light-tailed Gumbel class (Gumbel, 1958), and (iii) a negative shape parameter ($\xi < 0$) indicates a short-tailed or (bounded) negative-Weibull distribution (Weibull, 1951). This parameter is crucial for assessing the risk of rare floods and informing the design infrastructure to withstand such extremes. ..."

**5. Line 293: '…estimate the GEV parameters in a non-stationary context' – can you elaborate a bit on what exactly this refers to – is it about fitting multiple GEVs to different periods of the data to represent non-stationary conditions?**

We appreciate the reviewer's suggestion regarding the clarification of the wording "estimate the GEV parameters in a non-stationary context." This refers to allowing the GEV distribution parameters to vary with time, in order to capture temporal changes in the statistical behavior of annual peak flood time series. We have added this clarification in the revised manuscript, at page 15, lines 436-439: "... We have used the Generalized (Penalized) Maximum Likelihood Estimation (GMLE) method (Martins & Stedinger, 2000) to estimate the GEV parameters in a non-stationary context, by allowing the model parameters to vary with time (Coles, 2001). …"

.

---

## Author Comment (AC2)

**Reviewer #2**

This study performed a regional-scale assessment of climate change impacts on flood for Western Africa, using two large-scale hydrological models with the bias-corrected CMIP6 climate projections. I think the study has high potential to form valuable knowledge base of the likely changes in flood in Western Africa, but I have some major concerns on the approach taken in hydrologically modelling, which limited my capacity to assess the results presented. As such, I'd like to seek further clarification and justification from the authors on their chosen approach, or reconsideration of alternative approach, before proceeding to further review of the results.

We thank the reviewer for his encouraging feedback on the potential contribution of our study. We acknowledge the concern regarding the hydrological modelling approach. We have provided additional clarification and justification for our methodology in the revised manuscript.

**General comments:**

**1. Given the substantial lack of data in the study region, I'm wondering about the value of using rather complicated hydrological models (distributed and semi-physical) rather than simpler models (e.g., lumped conceptual models)? There is a lack of 1) motivation for exploring distributed and semi-physical modeling approach within the study objective (in the Introduction); 2) justification of modelling approach within Section 2.3 of the Materials and Methods. The start of the Introduction also touched on the challenge with data scarcity for the study region, which seems to suggest that uncertainties from input data might affect modelling (especially for more complex models which has higher data requirement) to some large degree - some assessments and/or discussion on this aspect would be useful.**

We agree that model complexity must be balanced against data availability, particularly in data-scarce regions such as West Africa. Nevertheless, our primary motivation for utilizing distributed models is to account for the spatial heterogeneity of runoff-generating processes such as variations in land use, soil properties, and rainfall patterns, which cannot be adequately captured by simple lumped models. It should also be noted that in these regions, many studies are based on simple models that rely exclusively on calibration, without explicitly taking basin properties into account. Our study proposes an important step forward by using process-based models, and in the future these models could also provide a better understanding of the complex interactions between climate and land-use changes. We have added in the introduction, pages 4-5, lines 127-138: "... Due to their simplicity and computational efficiency, lumped hydrological models have been widely applied in West Africa (Niel et al., 2003; Bodian et al., 2016; 2018; Kwakye & Bárdossy, 2020; Koubodana et al., 2021). However, because runoff generation is an inherently spatial and temporally dynamic process, changing environmental conditions may impact flood frequencies and water availability (Wilson et al., 1979; Haddeland et al., 2002; Descroix et al., 2018). Although lumped models often perform comparably or even better than distributed models at the catchment outlet (Reed et al., 2004), their main limitation lies in evaluating the overall catchment response simply at the outlet, without accounting for the contributions of upstream individual sub-basins (Cunderlik, 2003; Pokhrel et al., 2008; Jajarmizad et al., 2012). The main advantage of distributed models is not necessarily a higher accuracy of runoff simulations at specific points (e.g., outlet or gauge stations), but rather their broader applicability and ability to simulate the impacts of spatially varying drivers and scenarios (Gebremeskel et al., 2005; Tang et al., 2007; Thielen et al., 2009; Chu et al., 2010; Tran et al., 2018). …"

.

**2. In the current analyses, the HMF-WA model has not been calibrated, while calibration for LISFLOOD seems to be done previously which are not part of this study. This attracts several major questions on the modelling approach:**

**- The disadvantage of not calibrating HMF-WA is clearly demonstrated in the results (Figure 2), the largely unsatisfactory performance of the model suggests that we have low confidence that it could even well represent the historical flood events. Although the results is accompanied by brief discussion on this issue that 'projections of climate change impacts on African hydrological trends were produced using...' – the decision to use an uncalibrated model is generally not standard in the international literature and require much more justification.**

We partly agree with the reviewer that using an uncalibrated model is uncommon in the international literature to provide hydrological scenarios of climate change impacts. One of the most striking examples is the use of ISIMIP (Frieler et al., 2017) simulations (large-scale global models that are mostly uncalibrated) for numerous hydrological impact studies, including in the journal Science (Gudmundsson et al. 2021). Furthermore, most land-surface models used to provide hydrological scenarios are also not calibrated.

One of the core objectives of this study is to evaluate the consistency of climate change signals across hydrological models that differ in structure and complexity, and notably to identify whether the use of a calibrated model could provide different hydrological projections under climate scenarios. By including both a calibrated model (LISFLOOD) and an uncalibrated model (HMF-WA), we aim to investigate how model calibration influences the projection of flood trends under changing climatic conditions, and to evaluate the potential of uncalibrated hydrological models as a practical alternative for addressing data scarcity in poorly gauged regions for climate impact studies. We have added a justification in the description of selected hydrological models (Section 2.3, lines 312-319): "... Nevertheless, while calibration can enhance the accuracy of discharge simulations, several studies have highlighted that uncalibrated global hydrological models often exhibit comparable sensitivity to climate variability as the regional calibrated hydrological models, particularly when assessing relative changes in extreme events between future and historical periods (Gosling et al., 2017; Zhao et al., 2025). Therefore, whether a calibrated hydrological model offers different climate change projections than an uncalibrated model needs further investigation (Pechlivanidis et al., 2017). …". Moreover, our findings (line 803 in the revised manuscript) suggest that: "... using both models, the climate forcing has more importance than the hydrological representation itself."

**- On the LISFLOOD model, further justifications and details are required on the calibration process, including the input data, objective function, and cross-validation (if any). Such information on calibration is necessary for the reviewers/readers to assess the suitability of these models for the purpose of the study.**

We thank the reviewer for raising this important point. We have added a detailed description of the calibration process of the LISFLOOD model in the revised manuscript, at page 10, lines 291-319): "... The LISFLOOD version used in this study (OS LISFLOOD v4.1.3) was regionally calibrated with a 0.05° (~5 km) resolution, using in-situ discharge gauge stations with at least four years of daily measurements recorded after 1 January 1982. In this setup, model parameters are linked to global geospatial datasets describing catchment morphology and river networks, land use, vegetation characteristics, soil properties, lake distribution, and water demand (Salamon et al., 2024; Choulga et al., 2024). The Distributed Evolutionary Algorithms in Python (DEAP; Fortin et al., 2012) framework

.

was applied to optimize parameters in gauged catchments, with the modified Kling-Gupta Efficiency (KGE; (Gupta et al., 2009) utilized as the objective function. Calibration was performed over a continuous simulation period using ERA5 reanalysis meteorological forcing. Due to the varying length and temporal coverage of the discharge records used for calibration, model performance was assessed using all available observational data at each station, rather than splitting the records into separate calibration and validation periods. The LISFLOOD calibration tool is freely available at https://github.com/ec-jrc/lisflood-calibration."

**- Further, it is not clear whether the LISFLOOD models have been explicitly calibrated/evaluated to a flood context. Please see an example of calibration of hydrological models tailored to rarer floods, would the models used in this study benefit from a flood-centered calibration? Wasko et al., 2023. https://doi.org/10.1016/j.jhydrol.2023.129403**

We thank the reviewer for the relevant reference. Unlike in Wasko et al. (2023), where the GR4J lumped rainfall-runoff model was locally calibrated to rare floods using a flood-centered objective function, the LISFLOOD model calibration relies on the modified Kling-Gupta Efficiency (KGE; Gupta et al., 2009), which is not explicitly designed to prioritize rare or extreme events. Nevertheless, LISFLOOD has demonstrated robust performance in simulating daily river discharge across a large number of calibration sites worldwide, with a global median KGE of 0.70 (https://confluence.ecmwf.int/display/CEMS/GloFAS+v4+calibration+hydrological+model+performance). Importantly, LISFLOOD is the core hydrological model used in both the Global Flood Awareness System (GLOFAS) which provides an overview on upcoming floods in large world river basins (Alfieri et al., 2013; Hirpa et al., 2018; Harrigan et al., 2020; Prudhomme et al., 2024; Silva Peixoto et al., 2024) and the European Flood Awareness System (EFAS; Thielen et al., 2009; Matthews et al., 2024) which operates on a pan-European scale to provide short-to medium-range flood forecasts (Smith et al., 2016; Zábori et al., 2024), under the Copernicus Emergency Management Service (CEMS). Its proven applicability to large-scale hydrological forecasting and flood monitoring confirms its suitability for simulating floods, especially in large river basins. This is a key reason for its use in the present study. In addition, LISFLOOD GloFAS and EFAS set-ups are also used by the CEMS Global and European Drought Observatories (GDO, EDO, respectively) for low flow index and soil moisture anomaly estimation (e.g. Toreti et al., 2025). LISFLOOD GloFAS set-up also proved to enable adequate assessment of total water storage (e.g. Jensen et al., 2025) and is used for various purposes. Therefore, a flood-centered objective function was not used in the calibration process.

Moreover, we have assessed the LISFLOOD model's ability to simulate flood behavior, by comparing the distributions of observed and simulated annual maximum flows using the Anderson-Darling (AD) test at the 0.05 significance level (Scholz & Stephens, 1986). The results indicate that LISFLOOD reproduces the statistical properties of extreme flows reasonably well at a majority of the gauged stations, providing further confidence in its application for flood frequency analysis under historical and projected climate conditions. We have added this point into the discussion of the hydrological model evaluation results in the revised manuscript (lines 635-639): "... In addition, the satisfactory performance of the LISFLOOD model indicates that, although a flood-centered calibration approach could potentially improve its ability to capture extreme flows and their trends (Wasko et al., 2021), the current model setup provides a satisfactory basis for regional-scale flood trend assessments. …"

**3. Section 2.2 on data: given the substantial challenges in data availability for the study region, I think specific attention should be paid to the representativeness of the data to ensure they are not**

.

**biased towards a specific type of catchment, and/or specific time periods. I think this can be achieved by adding the following details:**

**- A summary table (possibly in the Supplementary) of the selected study sites, with information on their catchment areas, mean annual catchment-averaged rainfall, mean annual streamflow, and the range of years over which streamflow data is available.**

We thank the reviewer for this helpful suggestion. We have compiled a summary table of the selected gauge stations, providing key characteristics including catchment area, mean annual catchment-averaged rainfall, mean annual streamflow, and the range of years for which streamflow data are available. This table has been added to the Supplementary Material (Table S1).

**- The study site selection criteria mentioned 'a minimum of 10 years streamflow time series between 1950 and 2018' – does this allow for data gaps (i.e., days with missing or low-quality streamflow data), and if so, what is the maximum length of gaps allowed?**

As stated, our selection criterion required a minimum of 10 years of continuous available streamflow data between 1950 and 2018. This criterion allows for data gaps provided that there are no missing values near the potential annual peak flood). We have clarified this aspect in the revised manuscript (section 2.2, lines 213-218): "... To address the challenges associated with missing data in the database, we conducted a year-by-year visual inspection of hydrographs at each station as illustrated by Supplementary Figure S2. Years with data gaps near the flood peak were excluded from the analysis to avoid the risk of missing the true annual peak flood (Wilcox et al., 2018). Through this screening process, we ensured that no AMF values were derived from periods characterized by a lot of missing data."

[Figure]

Figure S2: Illustration showing the handling of missing data in an annual hydrograph of daily discharge measurements. A significant portion of data, particularly around the peak discharge period, is missing (highlighted by the red circle). Such a year is excluded from the analysis to ensure the accuracy of the annual peak flood sampling.

.

**- How variable is the land uses in this study region? If they are rather heterogeneous, a summary of key land use types in each catchment would also be useful.**

The study area is characterized by a heterogeneous landscape across the different catchments, with considerable variability in land use. We have added to the Supplementary Material (Table S2) a summary of key land cover types (forest, urban, crop, irrigated crops, grass, shrub, sparse, and bare) in each catchment, detailing the dominant land uses and their respective proportions.

Table S2: Land use distribution across different catchments in the study area. The table shows the proportion of each land use type (Forest, Urban, Crop, Irrigated Crops, Grass, Shrub, Sparse, and Bare) within the catchments identified by their unique IDs. Each value represents the proportion (percentage) of the respective land use type within a given catchment.

| ID | Forest | Urban | Crop | Crop Irrig | Grass | Shrub | Sparse | Bare |
|---|---|---|---|---|---|---|---|---|
| ADHI_114 | 0.06 | 0 | 0.46 | 0 | 0 | 0.48 | 0 | 0 |
| ADHI_121 | 0.27 | 0 | 0.29 | 0 | 0 | 0.44 | 0 | 0 |
| ADHI_123 | 0.83 | 0 | 0.11 | 0 | 0 | 0.06 | 0 | 0 |
| ADHI_131 | 0.74 | 0 | 0.17 | 0 | 0 | 0.09 | 0 | 0 |
| ADHI_144 | 0.49 | 0 | 0.49 | 0 | 0 | 0.02 | 0 | 0 |
| ADHI_163 | 0.96 | 0 | 0.01 | 0 | 0 | 0.03 | 0 | 0 |
| ADHI_172 | 0.05 | 0 | 0.78 | 0.01 | 0.02 | 0.14 | 0 | 0 |
| ADHI_179 | 0.27 | 0.01 | 0.72 | 0 | 0 | 0 | 0 | 0 |
| ADHI_180 | 0.31 | 0 | 0.67 | 0 | 0 | 0 | 0 | 0 |
| ADHI_183 | 0.56 | 0 | 0.44 | 0 | 0 | 0 | 0 | 0 |
| ADHI_187 | 0.33 | 0 | 0.39 | 0 | 0 | 0.27 | 0 | 0 |
| ADHI_198 | 0.52 | 0 | 0.25 | 0 | 0 | 0.23 | 0 | 0 |
| ADHI_270 | 0.76 | 0 | 0.17 | 0 | 0 | 0.07 | 0 | 0 |
| ADHI_276 | 0.74 | 0 | 0.14 | 0 | 0 | 0.12 | 0 | 0 |
| ADHI_304 | 0.02 | 0 | 0.04 | 0.03 | 0.25 | 0.37 | 0.26 | 0.07 |
| ADHI_315 | 0.05 | 0 | 0.74 | 0 | 0 | 0.2 | 0 | 0 |
| ADHI_316 | 0.09 | 0 | 0.41 | 0.01 | 0.11 | 0.08 | 0.05 | 0.25 |
| ADHI_319 | 0.14 | 0 | 0.38 | 0.01 | 0.09 | 0.11 | 0.03 | 0.25 |
| ADHI_320 | 0 | 0 | 0.62 | 0.02 | 0.21 | 0 | 0.12 | 0.03 |
| ADHI_321 | 0.33 | 0 | 0.51 | 0 | 0.01 | 0.13 | 0 | 0 |
| ADHI_324 | 0.37 | 0 | 0.48 | 0.01 | 0 | 0.14 | 0 | 0 |
| ADHI_325 | 0.83 | 0 | 0.09 | 0 | 0.04 | 0.03 | 0 | 0 |
| ADHI_332 | 0.39 | 0 | 0.02 | 0 | 0.58 | 0.01 | 0 | 0 |
| ADHI_372 | 0.02 | 0 | 0.8 | 0 | 0 | 0.17 | 0 | 0 |
| ADHI_390 | 0.7 | 0 | 0.23 | 0 | 0 | 0.07 | 0 | 0 |
| ADHI_394 | 0.62 | 0 | 0.33 | 0 | 0 | 0.04 | 0 | 0 |
| ADHI_507 | 0.58 | 0 | 0.18 | 0 | 0 | 0.24 | 0 | 0 |
| ADHI_510 | 0.8 | 0 | 0.08 | 0 | 0 | 0.11 | 0 | 0 |
| ADHI_511 | 0.43 | 0 | 0.44 | 0 | 0 | 0.13 | 0 | 0 |
| ADHI_515 | 0.32 | 0 | 0.54 | 0 | 0 | 0.14 | 0 | 0 |
| ADHI_519 | 0.05 | 0 | 0.74 | 0 | 0.09 | 0.11 | 0 | 0 |
| ADHI_531 | 0.5 | 0 | 0.44 | 0 | 0 | 0.05 | 0 | 0 |
| ADHI_548 | 0.78 | 0 | 0.01 | 0 | 0 | 0.2 | 0 | 0 |
| ADHI_550 | 0 | 0 | 0.21 | 0.01 | 0.69 | 0.01 | 0.05 | 0.03 |
| ADHI_560 | 0.08 | 0 | 0.38 | 0.02 | 0.23 | 0.2 | 0.04 | 0.06 |
| ADHI_571 | 0.11 | 0 | 0.49 | 0 | 0 | 0.39 | 0 | 0 |
| ADHI_585 | 0.63 | 0 | 0.29 | 0 | 0 | 0.07 | 0 | 0 |
| ADHI_587 | 0.56 | 0 | 0.23 | 0 | 0 | 0.21 | 0 | 0 |
| ADHI_592 | 0.1 | 0.01 | 0.87 | 0 | 0 | 0.02 | 0 | 0 |

.

| | | | | | | | | |
|---|---|---|---|---|---|---|---|---|
| ADHI_595 | 0.19 | 0 | 0.58 | 0.01 | 0.01 | 0.21 | 0 | 0 |
| ADHI_596 | 0.75 | 0 | 0.15 | 0 | 0 | 0.09 | 0 | 0 |
| ADHI_597 | 0.96 | 0 | 0.01 | 0 | 0 | 0.03 | 0 | 0 |
| ADHI_605 | 0.02 | 0 | 0.93 | 0.01 | 0 | 0.05 | 0 | 0 |
| ADHI_607 | 0.43 | 0 | 0.09 | 0 | 0 | 0.48 | 0 | 0 |
| ADHI_612 | 0.39 | 0 | 0.09 | 0 | 0 | 0.52 | 0 | 0 |
| ADHI_613 | 0.71 | 0 | 0.07 | 0 | 0 | 0.21 | 0 | 0 |
| ADHI_617 | 0.61 | 0 | 0.21 | 0 | 0 | 0.18 | 0 | 0 |
| ADHI_639 | 0.41 | 0 | 0.55 | 0 | 0 | 0.04 | 0 | 0 |
| ADHI_640 | 0.3 | 0 | 0.57 | 0.01 | 0.01 | 0.11 | 0 | 0 |
| ADHI_649 | 0.91 | 0 | 0.07 | 0 | 0 | 0.02 | 0 | 0 |
| ADHI_650 | 0.92 | 0 | 0.01 | 0 | 0 | 0.07 | 0 | 0 |
| ADHI_651 | 0.72 | 0 | 0.16 | 0 | 0 | 0.11 | 0 | 0 |
| ADHI_678 | 0.76 | 0 | 0.1 | 0 | 0 | 0.14 | 0 | 0 |
| ADHI_692 | 0.1 | 0 | 0.45 | 0.03 | 0.19 | 0.12 | 0.04 | 0.08 |
| ADHI_1183 | 0.19 | 0 | 0.4 | 0 | 0 | 0.4 | 0 | 0 |
| ADHI_1269 | 0.71 | 0.01 | 0.28 | 0 | 0 | 0.01 | 0 | 0 |
| ADHI_1400 | 0.07 | 0 | 0.8 | 0 | 0 | 0.13 | 0 | 0 |
| ADHI_1401 | 0.1 | 0 | 0.75 | 0 | 0 | 0.15 | 0 | 0 |

**- Sources of input data for the hydrological models e.g., rainfall, temperature – it is unclear where they are from, it is implied from the later Section 2.5 that rainfall and temperature were from GCM rather than observed, but it would be helpful to clarify this earlier in the data section.**

We appreciate the helpful suggestion from the reviewer. In response, we have revised Section 2.2 (lines 201–226) to clarify the sources of input data for the hydrological models. Now Section 2.2 reads: "**2.2 Observational data and climate forcings for hydrological experiments:** Daily streamflow data for the period 1950-2018 were obtained from the African Database of Hydrometric Indices (ADHI) recently developed by Tramblay et al. (2021). This database provides hydrometric indices computed from different data sources, with daily discharge time series that span at least 10 years. In the ADHI database, the size of the 441 West African catchments ranges from 95 to 2,150,000 km², and some stations have daily discharge data spanning over 44 years. Figure 1 shows the spatial distribution of the ADHI stations used in this study. We only selected watersheds from the ADHI database that met the following three criteria: (i) low regulation, determined through visual inspection of dam locations relative to watershed outlets (see Supplementary Figure S1), combined with a year-by-year analysis of annual hydrographs to assess the impact of dam operations on streamflow, (ii) surface area of less than 150,000 km², and (iii) a daily streamflow time series covering a minimum of 10 years between the 1950 and 2018. To address the challenges associated with missing data in the database, we conducted a visual inspection of hydrographs at each station as illustrated by Supplementary Figure S2. Years with data gaps near the flood peak were excluded from the analysis to avoid the risk of missing the true annual peak flood (Wilcox et al., 2018). Through this careful screening process, we ensured that no AMF values were derived from periods characterized by a lot of missing data. It is important to note that the observational streamflow data are not used to calibrate or drive the hydrological models. Instead, these observations serve as an independent benchmark to evaluate the ability of the hydrological models to reproduce key flood statistics during the historical period. The LISFLOOD model was calibrated using the ERA5 reanalysis dataset, which provides consistent and high-resolution precipitation and temperature fields. Moreover, ERA5 was also used as a reference for the bias correction of the five climate models from the CMIP6 ensemble that were used to drive the hydrological simulations for both the historical and future periods (see Section 2.4)."

.

**4. Section 2.3: I understand that the details of the two hydrological models are presented in the corresponding papers cited, but I think the readers could benefit from some additional background on these models, at least covering the key processes represented in each model on converting rainfall to runoff. This information is currently only partly available for the HWF-WA model (with only the recently added process representations listed) and not communicated for the LISFLOOD model. After presenting these, I'd also love to see a quick summary of the key differences between the models to justify your point in the Abstract that the two models 'differ in their hydrological process representation'.**

We appreciate the reviewer's suggestion. We have expanded Section 2.3 (lines 261-319) to clarify the key hydrological processes in each model and the key differences between the two models. The updated section now reads: "The HMF-WA model is adapted from the modular HMF model, and is designed for large-scale applications across West Africa (Rameshwaran et al., 2021). It employs a vertically integrated soil moisture scheme to simulate runoff production, driven by rainfall and potential evaporation inputs. Runoff generation considers soil drainage and a spatial probability distribution of soil moisture. Routing is based on a kinematic wave approach (Bell et al., 2007), with parallel pathways for surface and subsurface flow. Key enhancements over the classical HMF model include modules to simulate wetland inundation, endorheic basins, and anthropogenic water withdrawals, making it well-suited for semi-arid environments with complex hydrology (Rameshwaran et al., 2021). HMF-WA simulates spatially consistent river flows across West Africa at a $0.1° \times 0.1°$ spatial resolution. Although it has not yet been specifically calibrated to individual West African catchments using observed streamflow data where the model hydrology is configured to local conditions using spatial datasets of physical and soil properties, HMF-WA model evaluation against observational data indicates that it performs reasonably well in simulating both daily high and low river flows across most catchments. The median values of NSE (Nash-Sutcliffe efficiency), NSElog, and BIAS are 0.62, 0.82, and 0.06 (6 %), respectively (Rameshwaran et al., 2021).

The LISFLOOD model, developed by the Joint Research Centre (JRC) of the European Commission (https://ec-jrc.github.io/lisflood/), is a physical, spatially distributed hydrological model, designed for simulating several hydrological processes that occur in a catchment (Van Der Knijff et al., 2010). The LISFLOOD model simulates water processes using a three-layer soil water balance, along with groundwater and subsurface flow models. It accounts for several processes such as snow accumulation/melt, infiltration, evapotranspiration, groundwater flow, surface runoff, etc. Moreover, it supports the integration of human influences such as reservoirs and water abstraction. The numerical LISFLOOD simulation is driven by meteorological forcing (precipitation, temperature, and evapotranspiration) combined with high-resolution spatial data on terrain morphology, soil characteristics, land use, and water demand. This integrated setup allows the model to simulate runoff processes under diverse climatic and socio-economic conditions, capturing both natural and anthropogenic influences across heterogeneous landscapes. The runoff produced at every grid cell within the model domain is routed through the river network using a kinematic wave approach. The LISFLOOD version used in this study (OS LISFLOOD v4.1.3) was regionally calibrated with a 0.05° (~5 km) resolution, using in-situ discharge gauge stations with at least four years of daily measurements recorded after 1 January 1982. In this setup, model parameters are linked to global geospatial datasets describing catchment morphology and river networks, land use, vegetation characteristics, soil properties, lake distribution, and water demand (Salamon et al., 2024; Choulga et al., 2024). The Distributed Evolutionary Algorithms in Python (DEAP; Fortin et al., 2012) framework was applied to optimize parameters in gauged catchments, with the modified Kling-Gupta Efficiency (KGE; Gupta et al., 2009) utilized as the objective function. Calibration was performed over a continuous simulation

.

period using ERA5 reanalysis meteorological forcing. Due to the varying length and temporal coverage of the discharge records used for calibration, model performance was assessed using all available observational data at each station, rather than splitting the records into separate calibration and validation periods. The LISFLOOD calibration tool is freely available at https://github.com/ec-jrc/lisflood-calibration.

Globally, while both models use a kinematic wave routing scheme, HMF-WA and LISFLOOD differ significantly in their hydrological process representation. HMF-WA applies a vertically integrated soil moisture scheme with simplified runoff generation based on spatial soil moisture distribution. In contrast, LISFLOOD features a more detailed, physically-based three-layer soil model with an explicit representation of groundwater, snow processes, and anthropogenic influences. Furthermore, LISFLOOD has been calibrated using in-situ discharge data. Nevertheless, while calibration can enhance the accuracy of discharge simulations, several studies have highlighted that uncalibrated global hydrological models often exhibit comparable sensitivity to climate variability as the regional calibrated hydrological models, particularly when assessing relative changes in extreme events between future and historical periods (Gosling et al., 2017; Zhao et al., 2025). Therefore, whether a calibrated hydrological model offers different climate change projections than an uncalibrated model needs further investigation (Pechlivanidis et al., 2017).”

**Specific comments:**

**1. Line 51 – it will be clearer if the change in flood magnitude can be summarized specific to the flood return period(s) investigated.**

We appreciate the reviewer's suggestion. We now specify the change in flood magnitude by return period and future horizon in the abstract (lines 52-55): “... Flood magnitudes are projected to increase at 94% (96%) of stations for the 2-year (20-year) event in the near-term future, and at 88% (93%) of stations for the 2-year (20-year) event in the long-term future, with some locations expected to experience increases exceeding 45%. …”

**2. Figure 1 caption: ‘grey lines’ instead of ‘white lines’?**

We have changed the word “white” by “grey” in the wording caption of Figure 1 (page 8): “Figure 1: Spatial distribution of the ADHI stations used in this study, covering the three climatic zones in the West African region, as delimited by the blue isohyets (600 mm and 1200 mm annual rainfall) on the map. The color ramp of the circles indicates the record lengths of flood data (in years). The blue lines represent isohyets delimiting West African climatic regions, and the grey lines indicate the borders of West African countries.”

**3. Line 177 – decision on ‘low regulation’ catchments: the Supplementary Fig. 1 suggested that this is based on whether there is a dam located near the watershed outlet, with no information what defines a dam ‘near’ or ‘far from’ the outlet – was this based on visual inspection, or a threshold distance used? If the latter, how was the threshold distance determined?**

We appreciate the reviewer’s constructive comment regarding the decision on ‘low regulation’ catchments. The identification of "low regulation" catchments was based on visual inspection of the dam locations relative to the watershed outlet, using both the GRanD database (https://www.globaldamwatch.org/grand) and Google maps, combined with a year-by-year analysis of the annual hydrographs. This allowed us to verify whether the dam's construction or operational start

.

date caused noticeable changes in the streamflow regime. No fixed distance threshold was applied. We have clarified this aspect in the description of the observational data, Section 2.2, lines 208-213: "... We only selected watersheds from the ADHI database that met the following three criteria: (i) low regulation, determined through visual inspection of dam locations relative to watershed outlets (see Supplementary Figure S1), combined with a year-by-year analysis of annual hydrographs to assess the impact of dam operations on streamflow, (ii) surface area of less than 150,000 km², and (iii) a daily streamflow time series covering a minimum of 10 years between the 1950 and 2018. …"

**4. Section 2.6.1 – the introductory section for GEV is very informative, however, I think it could benefit from additional information on what positive and negative shape parameters mean, which seem to be useful context to the subsequent discussion on the plausible values of the shape parameter.**

We thank the reviewer for the insightful suggestion. We have now added information about the meaning of shape parameter values, in Section 2.6.1, lines 418-426: "... The shape parameter ($\xi$) governs the tail behaviour of the GEV distribution, which encompasses three types of extreme value distributions (Coles, 2001): (i) a positive shape parameter ($\xi > 0$) indicates a heavy-tailed Fréchet case (Fréchet, 1927), suggesting an increased probability of extreme flooding events, (ii) a null shape parameter ($\xi = 0$) suggests a light-tailed Gumbel class (Gumbel, 1958), and (iii) a negative shape parameter ($\xi < 0$) indicates a short-tailed or (bounded) negative-Weibull distribution (Weibull, 1951). This parameter is crucial for assessing the risk of rare floods and informing the design infrastructure to withstand such extremes. ..."

**5. Line 293: '…estimate the GEV parameters in a non-stationary context' – can you elaborate a bit on what exactly this refers to – is it about fitting multiple GEVs to different periods of the data to represent non-stationary conditions?**

We appreciate the reviewer's suggestion regarding the clarification of the wording "estimate the GEV parameters in a non-stationary context." This refers to allowing the GEV distribution parameters to vary with time, in order to capture temporal changes in the statistical behavior of annual peak flood time series. We have added this clarification in the revised manuscript, at page 15, lines 436-439: "... We have used the Generalized (Penalized) Maximum Likelihood Estimation (GMLE) method (Martins & Stedinger, 2000) to estimate the GEV parameters in a non-stationary context, by allowing the model parameters to vary with time (Coles, 2001). …"

.

.